# Trends and forecasts of leprosy for a hyperendemic city from Brazil's northeast: Evidence from an eleven-year time-series analysis

Antônio Carlos Vieira Ramos[1]*, Dulce Gomes[2], Marcelino Santos Neto[3], Thaís Zamboni Berra[1], Ivaneliza Simionato de Assis[4], Mellina Yamamura[5], Juliane de Almeida Crispim[1], José Francisco Martoreli Júnior[1], Alexandre Tadashi Inomata Bruce[1], Felipe Lima dos Santos[1], Ludmilla Leidianne Limirio Souza[1], Yan Mathias Alves[1], Hamilton Leandro Pinto de Andrade[1], Marcos Augusto Moraes Arcoverde[6], Flávia Meneguetti Pieri[7], Ricardo Alexandre Arcêncio[1]

1 Department of Maternal-Infant Nursing and Public Health, University of São Paulo at Ribeirão Preto College of Nursing, Ribeirão Preto, São Paulo, Brazil, 2 Department of Mathematics, University of Évora, Évora, Portugal, 3 Center for Social Sciences, Health and Technology, Federal University of Maranhão, Imperatriz, Maranhão, Brazil, 4 University Center Dinâmica of Cataratas, Foz do Iguaçu, Paraná, Brazil, 5 Nursing Department, Federal University of São Carlos, São Carlos, São Paulo, Brazil, 6 Center for Education, Letters and Health, Western Paraná State University, Campus Foz do Iguaçu, Foz do Iguaçu, Paraná, Brazil, 7 Department of Nursing, Londrina State University, Londrina, Paraná, Brazil

* antonio.vieiraramos@outlook.com

**Data Availability Statement:** The minimal de-identified dataset to replicate the study findings is available as a Supporting Information file.

## Abstract

This study's objective was to estimate the temporal trends of leprosy according to sex and age groups, as well as to estimate and predict the progression of the disease in a hyperendemic city located in the northeast of Brazil. This ecological time-series study was conducted in Imperatriz, Maranhão, Brazil. Leprosy cases diagnosed between 2006 and 2016 were included. Detection rates stratified by sex and age groups were estimated. The study of temporal trends was accomplished using the Seasonal-Trend Decomposition method and temporal modeling of detection rates using linear seasonal autoregressive integrated moving average model according to Box and Jenkins method. Trend forecasts were performed for the 2017–2020 period. A total of 3,212 cases of leprosy were identified, the average incidence among men aged between 30 and 59 years old was 201.55/100,000 inhabitants and among women in the same age group was 135.28/100,000 inhabitants. Detection rates in total and by sex presented a downward trend, though rates stratified according to sex and age presented a growing trend among men aged less than 15 years old and among women aged 60 years old or over. The final models selected in the time-series analysis show the forecasts of total detection rates and rates for men and women presented a downward trend for the 2017–2020 period. Even though the forecasts show a downward trend in Imperatriz, the city is unlikely to meet a significant decrease of the disease burden by 2020.

**Funding:** This study was financed in part by the Coordenação de Aperfeiçoamento de Pessoal de Nível Superior – Brasil (CAPES) – Finance Code 001. ACVR received financial assistance from the CAPES, Financing Code 001. Website: https://www.capes.gov.br/ The funders had no role in study design, data collection and analysis, decision to publish, or preparation of the manuscript.

**Competing interests:** The authors have declared that no competing interests exist.

## Introduction

Leprosy is an infectious disease caused by *Mycobacterium leprae*, which mainly affects the skin and peripheral nervous system, resulting in neuropathies and associated problems over the long term, including physical deformities and disabilities [1].

Even though leprosy has been eliminated as public health problem in many countries in the world (prevalence <1 case every 10,000 inhabitants) since the year 2000, leprosy still persists in developing countries as a serious public health problem [2, 3]. After the introduction of Multidrug Therapy (MDT) and the high vaccination coverage of the Bacillus Calmette-Guérin (BCG), especially in children, the burden of leprosy has decreased considerably worldwide. However, in some nations the elimination of the disease (zero transmission) and decreased detection of new cases continue to be important challenges for a world without leprosy [4].

In 2016, the World Health Organization (WHO) published the Global Leprosy Strategy 2016 −2020: Accelerating towards a leprosy-free world, the objectives are to decrease the disease's global and local burden, decrease the cases of children with deformities, decrease the new cases diagnosed with grade 2 physical disabilities to less than one case per 1 million inhabitants, and review all laws that somehow lead to the discrimination against people with leprosy [4].

The global detection rate for leprosy in 2018 was 1.93 cases/100,000 inhabitants and the countries that presented the highest rates were India, Brazil and Indonesia, responsible for 79.6% of the cases reported [5]. In the same year, Brazil presented a detection rate of new cases of 12.94 cases/100,000 inhabitants, accounting for 93% of the total cases reported in the Americas [5, 6]. Brazil has presented a downward trend in the number of cases in recent years, from 37,610 cases in 2009 to 28,660 in 2018 [5]; some regions, though, like the north, midwest and northeast still present high rates of the disease [7].

An ecological study using time-series analysis conducted in cities with a high risk for transmission of the disease located in Mato Grosso (midwest), Tocantins, Rondônia, Pará (north) and Maranhão (northeast), reports a decrease in the total detection rate from 89.10 to 56.98 cases/100,000 inhabitant between 2001 and 2012 [8]. According to the authors, there was a significant decrease in overall detection rates and among individuals younger than 15 years old in the study regions. However, the rate of new cases with grade 2 physical disability per 100,000 inhabitants remained stable over the period, suggesting late diagnosis and possibly underreporting of cases.

Following the WHO recommendations and the need to reduce the burden of the disease in Brazil, in 2016 the "*Diretrizes para vigilância, atenção e eliminação da hanseníase como problema de saúde pública*" [Guidelines for leprosy surveillance, care and elimination as a public health problem] was published, which discussed health promotion and health education actions, active case-finding for early detection and treatment, prevention and rehabilitation, and surveillance of those who had contact with the disease. These guidelines reaffirm the importance of adopting epidemiological indicators to monitor the progression of the disease, operational indicators to assess the quality of services and to include an indicator to verify the proportion of cases according to sex, reinforcing the importance of gender in the causality of the disease [9].

In addition to the fact that leprosy is a tropical disease that has been neglected, its association with poverty and social inequality, the disease presents a sex-specific distribution in terms of morbidity [10–12]; that is, men are more frequently affected than women in most regions of the world (including Brazil) [13]. However, in many countries, women are late diagnosed and have a higher proportion of degrees of physical disability, in addition to the fact that the stigma of the disease is greater in women [14]. Leprosy is a disease known for leading to different representations and effects between men and women, in different social contexts, and as a consequence, it accentuates gender inequalities from the Brazilian sociocultural point of view [15].

Studies addressing the temporal trend of leprosy cases were found in the literature [8, 10, 16], but few studies discuss the disease's temporal behavior according to sex. A study carried out in the state of Bahia (Northeast region of Brazil) by Souza et al. (2018) analyzed temporal trends in terms of sex and verified that the disease behaves differently according to sex: there is a tendency for decreased detection coefficients among women, but detection coefficients remain stable among men, though the results were not statistically significant [15].

No study was found among time-series studies that included future predictions of leprosy detection rates and that also considered these according to sex. Considering the previous discussion, this study's objective was to estimate the temporal trends of leprosy according to sex and age groups, as well as to estimate and predict the progression of the disease in a hyperendemic city located in the northeast of Brazil.

## Materials and methods

### Study setting and design

This ecological time-series study [17] was conducted in the city of Imperatriz, in the state of Maranhão, located in the Northeast of Brazil (Fig 1).

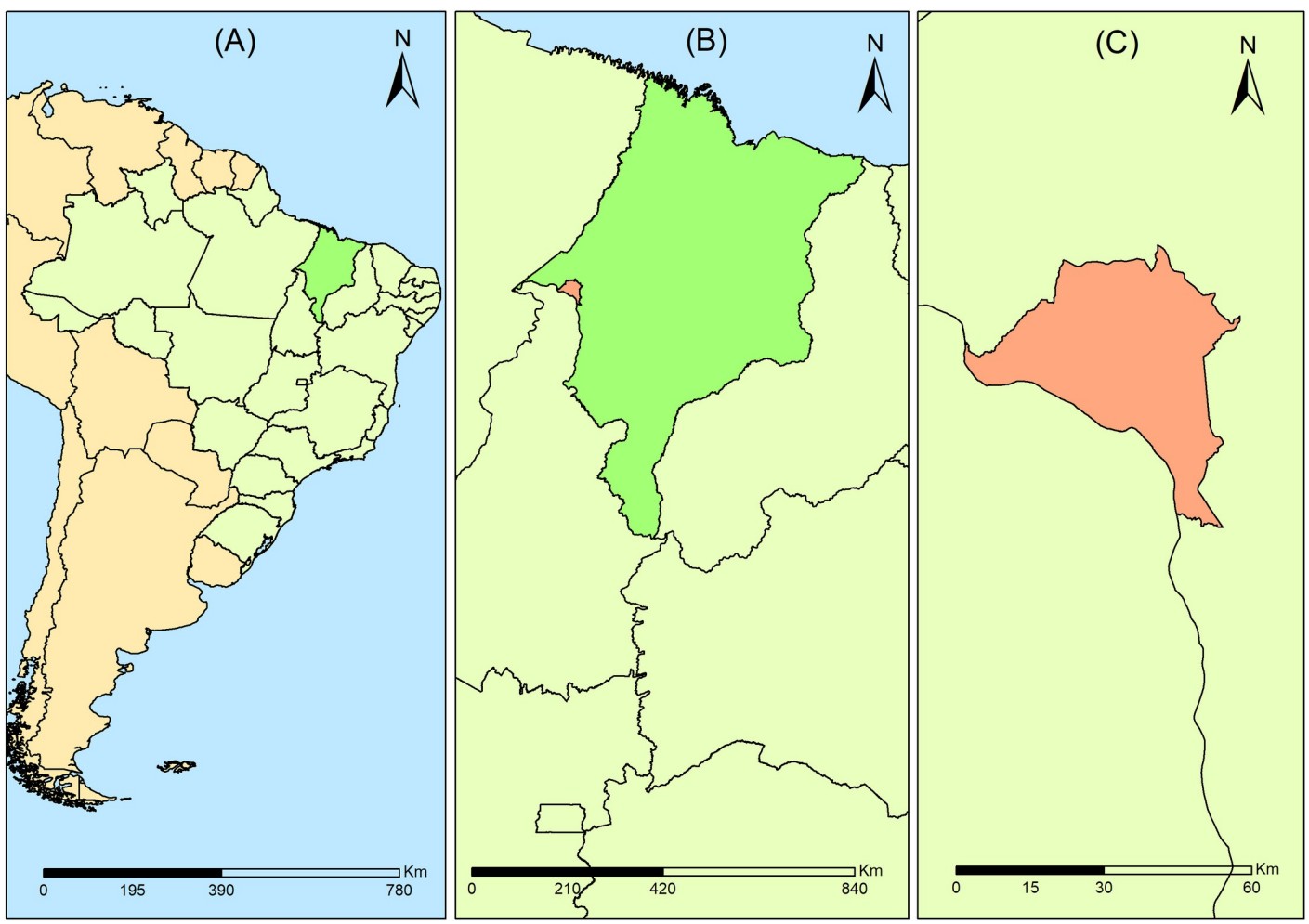

**Fig 1. Location map of the study setting, Imperatriz, MA, Brazil.** (A) Brazil; (B) State of Maranhão; (C) City of Imperatriz. Source: Authors.

Imperatriz is located 626 km from the capital of Maranhão, São Luís, and is the second largest city in the state and the 23rd largest city in the Brazilian northeast. According to the Demographic Census of the Brazilian Institute of Geography and Statistics (IBGE), in 2010, Imperatriz has a population of 247,505 inhabitants, with a demographic density of 180.79 inhabitants/km$^2$ with a territorial area of 1,368,988 km$^2$ [18]. In the same year, the main social indicators are: an illiteracy rate of 9.7%, Human Development Index (HDI) of 0.73 and Gini Index of 0.46. In terms of basic sanitation, 23% of the city has a sewage system and 86% has a drinking water supply [18, 19].

In 2016, the detection rate of new cases of leprosy in the state of Maranhão was 47.30 cases/ 100,000 inhabitants, classifying the state as the third most endemic in Brazil. In the same year, Imperatriz presented a detection rate of 62.23 cases/100,000 inhabitants, marking it as a Brazilian city with hyperendemicity levels [6, 9, 20].

## Study population and sources of information

All new leprosy cases reported to SINAN (Notifiable Diseases Information System) between 2006 and 2016 were selected. SINAN is the Brazilian information system responsible for recording and processing information regarding reportable diseases in the entire country, providing morbidity bulletins and reports. It is one of the main surveillance systems in Brazil [21].

The variables adopted in this study include date when leprosy cases were reported in the SINAN (notification date), age and sex. Data were collected at the health surveillance service from the city's regional management unit, state government of Maranhão in May 2018. During the collection process, data were tabulated in spreadsheets in Microsoft Office Excel® 2013, a process in which the database was validated and duplicated reports were removed.

After validating the database, detection rates were calculated per month (per 100,000 inhabitants). The calendar adjustment technique was applied in the calculation considering the number of days of each month in order to improve the representation of rates in the study period. After adjusting for the calendar effect, both the total detection rate and the detection rate stratified per sex (male and female) was calculated for four age groups (<15 years old; 15 to 29 years old; 30 to 59 years old; and ≥60 years old). The city's total population was considered in the computation of the total detection rate, while the population of men and women, with their respective age groups, was considered for the stratified detection rates. The size of the resident populations used as the denominator was based on the 2010 Census and the intercensal estimates (2006–2016) elaborated by the IBGE.

Leprosy detection rates were smoothed by the moving average technique, considering the average of three months (prior, current and posterior), in order to remove noise and better reveal the underlying causal process.

## Statistical analysis

An exploratory analysis of monthly leprosy detection rates (smoothed and with calendar adjustment correction) was performed according to sex and age group. Additionally, the Average Monthly Percentage Change (AMPC) of detection rates was calculated according to sex and age groups, identifying, in terms of average percentage, the rates of increase or decrease over the study period.

Afterwards, the progression of the disease trend was characterized according to sex and age using the robust Seasonal-Trend using Loess (STL) decomposition method by Cleveland et al. (1990) [22]. For that, at each point in time $t$, the time series $X_t$ is given by the sum of three components: seasonality ($S_t$), trend ($T_t$) and noise ($Z_t$). This decomposition is based on locally weighted regression (Loess) of the seasonality and trend components.

To model the monthly rates of total detection and detection by sex, as well as the forecast of respective trends, we used the linear seasonal autoregressive integrated moving average (ARIMA Seasonal) model and the usual Box and Jenkins method to choose the appropriate models based on the data structure itself [23].

The ARIMA Seasonal model–SARIMA *(p, d, q) (P, D, Q)$_S$*–enables describing the variability of time-related, linear, stationary ($d = D = 0$) or non-stationary (otherwise) processes and are written as follow:

$$\Delta(B^s)\Phi(B)(1 - B)^d(1 - B^s)^D T(X_t) = \Psi(B^s)\Theta(B)Z_t$$

Where:

$$\Phi(B) = 1 - \phi_1 B - \phi_2 B^2 - \cdots - \phi_p B^p, \ \Theta(B) = 1 - \theta_1 B - \theta_2 B^2 - \cdots - \theta_q B^q$$

respectively, the autoregressive and moving average polynomials of the non-seasonal part and,

$$\Delta(B^s) = 1 - \Phi_1 B - \Phi_2 B^2 - \cdots - \Phi_P B^P e\Psi(B^s) = 1 - \Theta_1 B - \Theta_2 B^2 - \cdots - \Theta_Q B^Q$$

respectively, autoregressive polynomials and moving average polynomials of the seasonal part of period *S*. *T* is the transformation to stabilize, if necessary, the variance (usually called Box-Cox transformation), while $Z_t$ represents the white noise process (uncorrelated process, null mean, and constant variance).

Letters *p* and *q* represent, respectively, the number of parameters of autoregressive parts and moving average parts, with the seasonal period of length *S*, and letters *P* and *Q* are the equivalent number of these parameters between the seasonal periods. Letters *d* and *D*, respectively, represent degrees of simple differentiation and the seasonal differentiation necessary to transform a non-stationary into a stationary series [24].

The maximum likelihood method was used to estimate the model's parameters. The usual tests of absence of autocorrelation (Portmanteau tests: Ljung-Box and Box-Pierce), randomness (Rank and Turning Point tests), and normality (Kolmogorov-Smirnov test) were used to validate the model, in the analysis of the residuals, along with a t-test for zero mean.

Whenever more than one model was appropriate, the best model was chosen considering the parsimony principle and the lowest values of Akaike information criterion (AIC) and Bayesian information criterion (BIC).

A set of tests was performed with data concerning the last two years (2015 and 2016) to assess the models' predictive performance. The following measures were considered to assess this predictive performance: Root Mean Square Error (RMSE), Mean Absolute Error (MAE) and Mean Absolute Percentage Error (MAPE), which allow assessing the precision of estimates or forecasts. According to a model's criteria, the most appropriate model will always be the one with the fewest errors [24]. Afterwards, data and tendency forecasts were performed for the four-year period (2017 to 2020).

The method proposed by Box and Jenkins consists of an interactive process composed of five stages: time series stationarization (through Box-Cox transformations followed by simple and/or seasonal differentiations); identification of the model and respective orders; estimation of parameters; model validation and prediction of future values [22–24].

All the analyses were performed using the RStudio® version 3.5.2 (https://rstudio.com).

## Ethical aspects

The study project was approved by the Institutional Review Board at the University of São Paulo, College of Nursing (EERP/USP) under Certificate of Presentation for Ethical

Appreciation (CAAE) No. 44637215.0.0000.5393. No consent forms were signed because only secondary data were used and the participants were not identified, as the data were analyzed anonymously.

## Results

A total of 3,212 leprosy cases were reported between 2006 and 2016 in Imperatriz. Table 1 presents the descriptive statistics of cases according to sex, age groups and AMPC, showing in absolute numbers, that the group aged between 30 and 59 years old predominated among total of cases (1566; rate = 166.46/100,000 inhabitants), men (892; rate = 201.55/100,000 inhabitants) and women (674; rate = 135.28/100,000 inhabitants). High rates were found among individuals aged 60 years old or over for total of cases (rate = 257.42/100,000 inhabitants), men (rate = 357.58/100,000 inhabitants) and women (rate = 173.70/100.000 inhabitants). The group of individuals younger than 15 years old also presented a large number of cases, with a rate of 40.26 cases/100,000 inhabitants among total of cases, a rate of 44.14 cases/100,000 inhabitants among men and 36.29/100,000 among women.

There is a continuous increase in the rates of case detection as the age groups increase, with the lowest rates being for children under 15 years old and the highest for those aged 60 years old or over.

In regard to AMPC trends, the male group younger than 15 years old (0.41%) and the female group aged 60 years old or over (0.09%) presented moderate growth in the study period.

Fig 2 presents the time trend for total detection (in black), among men (in red), and among women (in blue) distributed over the study period. In general terms, the three detection rates present the same decreasing behavior for the study period. Analyzing the comparison between the three trends, it is possible to observe stability in the period from 2014 to 2016.

**Table 1. Profile of leprosy cases according to sex, age group, and average percentage of rates, Imperatriz, MA, Brazil (2006–2016).**

| Total cases | | | |
|---|---|---|---|
| **Age groups (years)** | **Absolute frequency (3212)** | **Rates (100,000 inhabitants)** | **AMPC (%)** |
| <15 | 296 | 40.26 | -0.39 |
| 15 to 29 | 773 | 93.98 | -1.29 |
| 30 to 59 | 1566 | 166.46 | -0.59 |
| ≥60 | 577 | 257.42 | -0.66 |
| **Men** | | | |
| **Age groups (years)** | **Absolute frequency (1850)** | **Rates (100,000 inhabitants)** | **AMPC (%)** |
| <15 | 164 | 44.14 | 0.41 |
| 15 to 29 | 429 | 108.51 | -1.26 |
| 30 to 59 | 892 | 201.55 | -0.32 |
| ≥60 | 365 | 357.58 | -1.16 |
| **Women** | | | |
| **Age groups (years)** | **Absolute frequency (1362)** | **Rates (100,000 inhabitants)** | **AMPC (%)** |
| <15 | 132 | 36.29 | -0.52 |
| 15 to 29 | 344 | 80.53 | -1.04 |
| 30 to 59 | 674 | 135.28 | -0.78 |
| ≥60 | 212 | 173.70 | 0.09 |

AMPC, Average Monthly Percentage Change.
Source: Authors.

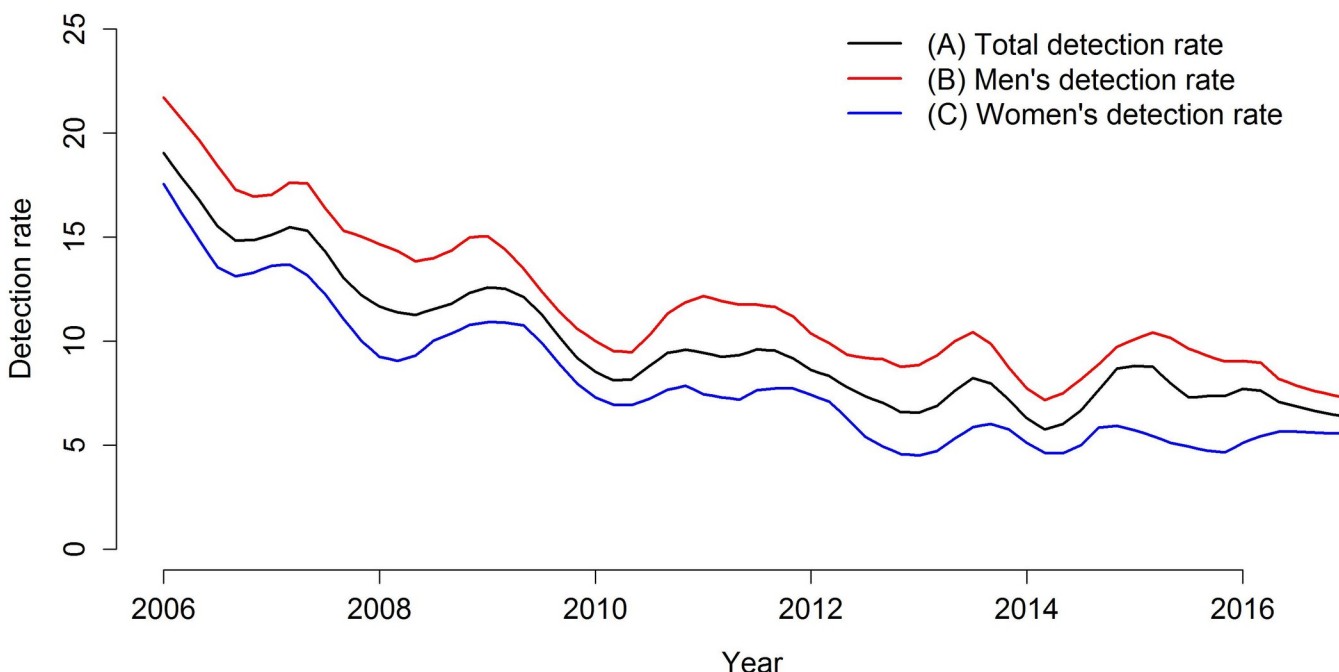

**Fig 2. Trend of leprosy total detection rates and detection rates among men and women, Imperatriz, MA, Brazil (2006–2016).** (A) Total detection rate; (B) Men's detection rate; (C) Women's detection rate.

The time series of the ratio between the detection rates of men and women is shown in Fig 3. The red line indicates the situation in which both rates would be equal (numerator equal to the denominator), and the blue line indicates the time period in that the ratio shows a change

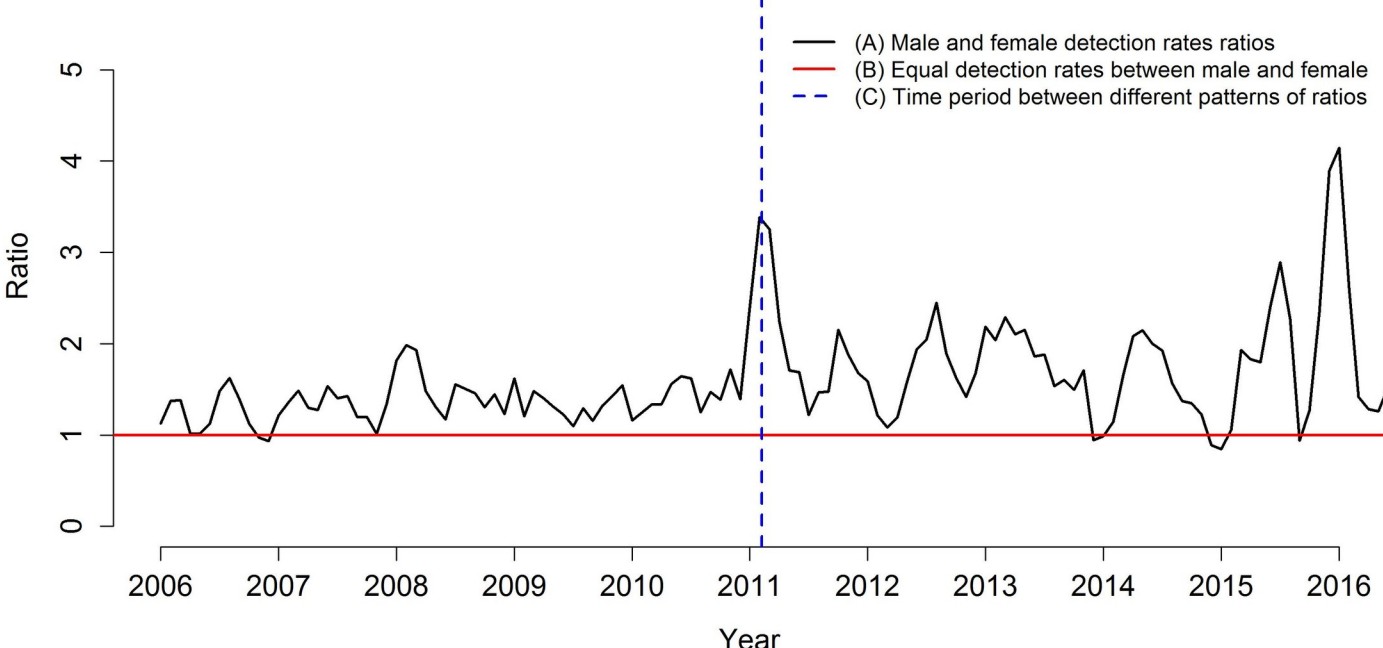

**Fig 3. Ratio between detection rates for men and women, Imperatriz, MA, Brazil (2006–2016).** (A) Male and female detection rates ratios; (B) Equal detection rates between male and female; (C) Time period between different patterns of ratios.

in behavior. Over time, the rate of men is generally higher than that of women, noting that, in the period from January 2006 to approximately September 2010, the rate of men reached, at most, double that of women; in January 2011 this difference exceeded the triple, and the quadruple in 2016.

Women showed slightly higher rates than men between September and December 2006, September 2013, between October 2014 and January 2015 and, finally, in July 2015.

Tendency toward leprosy according to sex and age (Fig 4) shows a decreasing trend for all age groups and sex, except for men younger than 15 years old and women aged 60 years old or over, reflecting specifically the AMPC by age and sex. Men under 15 years old showed decreasing trends from 2006 to 2014, after which they showed an increasing trend until the end of the study period. Women aged 60 years old or over showed a peak of detection between the years 2008 and 2009, with a decrease until 2011, and subsequently a continuous growth trend until the year 2016.

For women younger than 15 years old and aged between 15 to 29 years old there was a slight increase in the last year.

Despite the downward trend seen in the age group between 30 and 59 years old, both among women and men, high rates of disease were found in the entire study period.

As previously mentioned, the temporal modeling of leprosy detection rates, total and separated by sex, shows a downward trend, revealing that the series are not stationary. Thus, we

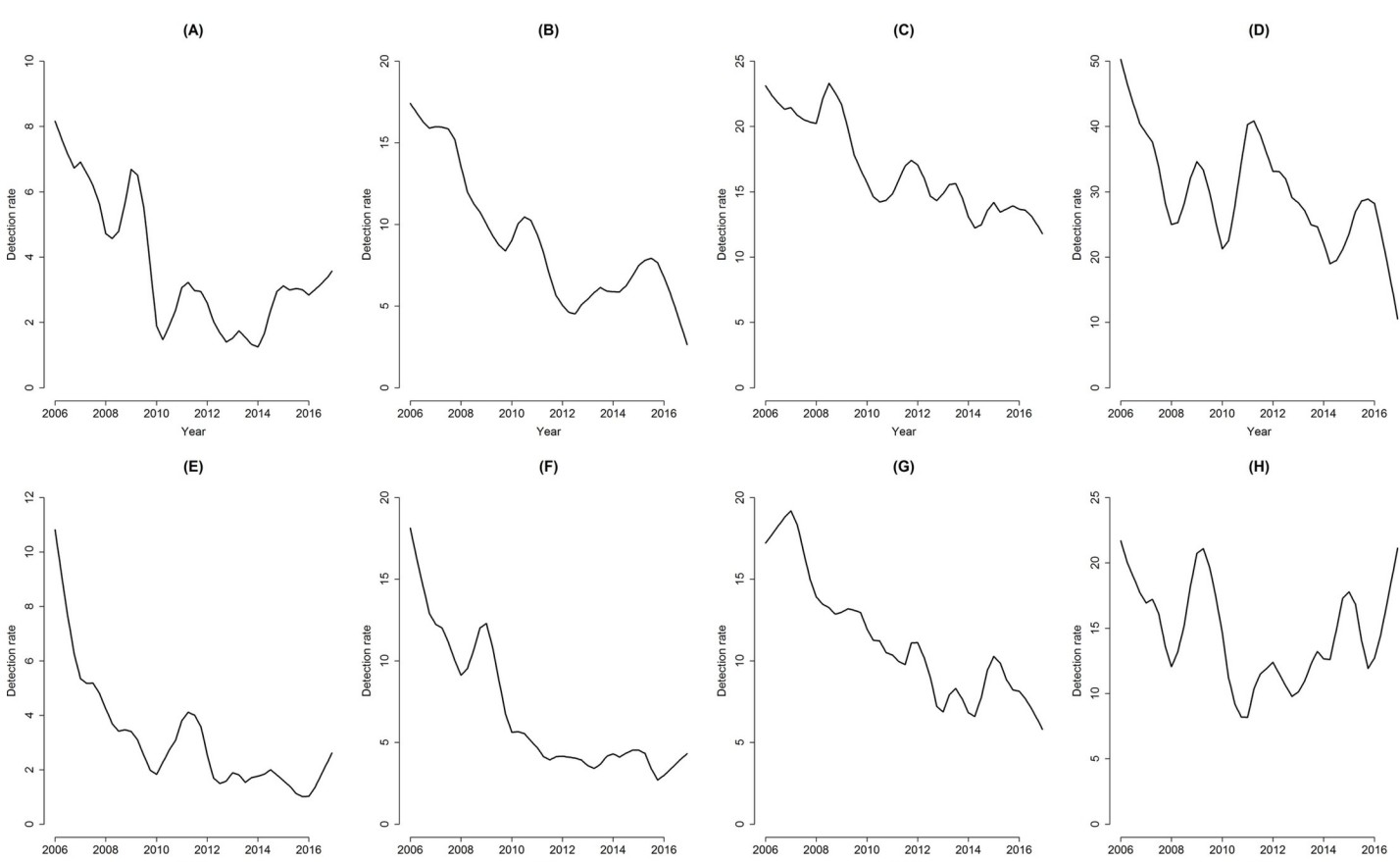

**Fig 4. Trends of leprosy according to sex and age groups, Imperatriz, MA, Brazil (2006–2016).** (A) Men younger than 15 years old; (B) Men aged between 15 and 29 years old; (C) Men aged between 30 and 59 years old; (D) Men aged 60 years old or over; (E) Women younger than 15 years old; (F) Women aged between 15 and 29 years old; (G) Women aged between 30 and 59 years old; (H) Women aged 60 years old or over.

**Table 2. Analysis of residuals of temporal modeling of detection rates, Imperatriz, MA, Brazil (2006–2016).**

| Tests | Total detection rate | | Detection rate for men | | Detection rate for women | |
|---|---|---|---|---|---|---|
| | Test statistics | P-value | Test statistics | P-value | Test statistics | P-value |
| Ljung-Box | 0.34 | 0.55 | 0.09 | 0.75 | 0.01 | 0.94 |
| Box-Pierce | 0.33 | 0.56 | 0.09 | 0.75 | 0.01 | 0.94 |
| Rank test | 0.33 | 0.74 | 0.57 | 0.56 | 0.96 | 0.33 |
| Turning Point | -1.38 | 0.16 | -0,55 | 0,57 | 0.27 | 0.78 |
| Kolmogorov–Smirnov | 0.06 | 0.64 | 0,05 | 0,88 | 0.08 | 0.29 |
| T test for the means | -0.01 | 0.29 | -0,00 | 0,98 | -0.00 | 0.91 |

Source: Authors.

performed Box-Cox transformations to stabilize variance and simple differentiations to stabilize the mean, transforming non-stationary series into stationary series. Some candidate models were chosen and their parameters were estimated when analyzing the autocorrelation and partial autocorrelation functions. After verifying the significance of the models' parameters, and considering the smallest AIC and BIC, the models that appear to be the most adequate in terms of ability to describe variability of data over time, as well as the forecasts' good performance, were: ARIMA (3,1,0) for the total detection rate, SARIMA (9,1,0) $(1,0,0)_{12}$ for detection rate among men, and ARIMA (0,1,3) for women.

The analysis the residual of the estimated models (Table 2) shows that all models are consistent with the models' assumptions (i.e., are independent and identically distributed, with normal distribution of zero mean, and constant variance).

The quality of forecasts was analyzed by comparing the set of tests (2015–2016), revealing very low precision measures (RMSE, MAE and MAPE), meaning that, on average, only 10.46% of the total detection rate of leprosy are incorrect (Table 3).

The modes' final adjustment, data forecasts, trends, and trend forecasts are presented in Figs 5–7. The figures show the adequate adjustment of three models, as well as its forecasts. Forecasts for total detection rates and rates for women and men indicate a downward trend for leprosy in the 2017–2020 period, with behavior very similar between sexes.

## Discussion

The present study identified a downward trend in the rate of detection of leprosy in the period from 2006 to 2016 indicating that such behavior will be maintained in the future, according to the models and time forecasts. Despite the decrease in the detection of the disease, Imperatriz will continue to present a high leprosy burden in the year 2020.

In 2016, Imperatriz presented a total detection rate of 62.23 cases/100,000 inhabitants, which, according to parameters provided by the Ministry of Health, classifies the city as

**Table 3. Predictive analysis of the detection rates models, Imperatriz, MA, Brazil (2006–2016).**

| Test | Total detection rate | Detection rate for men | Detection rate for women |
|---|---|---|---|
| RMSE | 1.23 | 1.34 | 1.25 |
| MAE | 0.95 | 1.06 | 0.94 |
| MAPE | 10.46 | 10.00 | 13.78 |

RMSE, Root Mean Square Error; MAE, Mean Absolute Error; MAPE, Mean Absolute Percentage Error (MAPE).
Source: Authors.

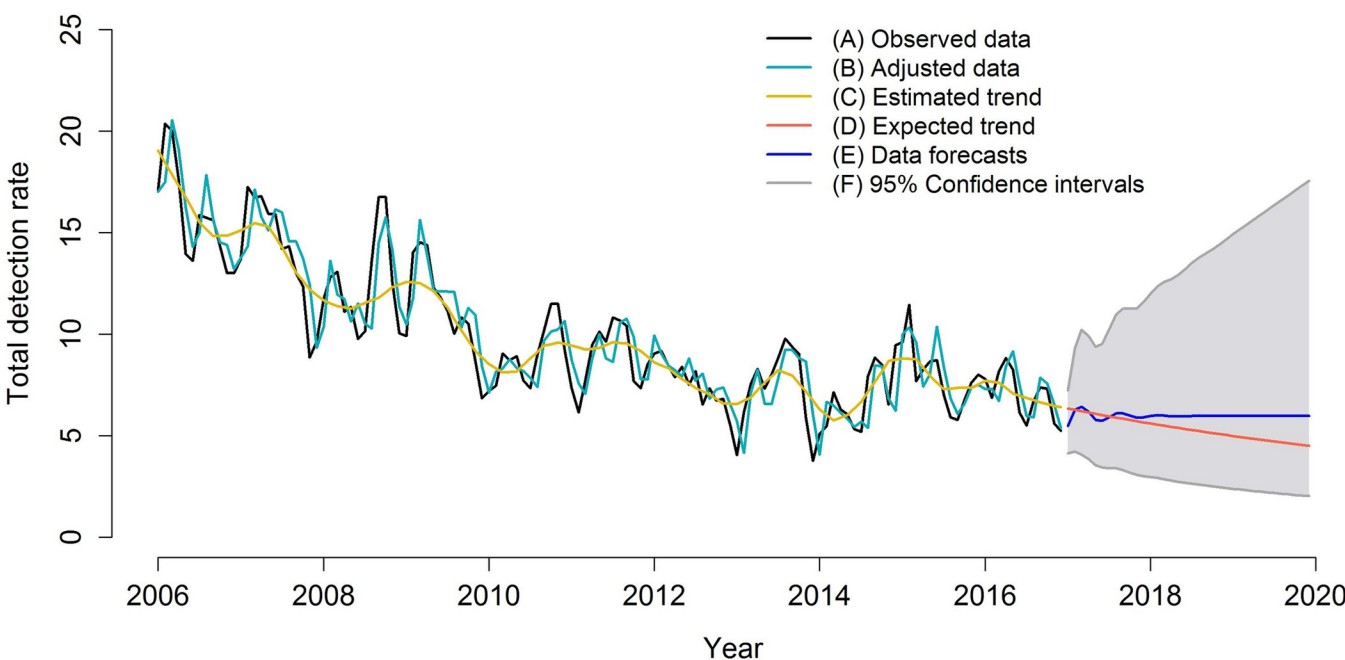

**Fig 5. ARIMA model (3,1,0) adjusted for leprosy total detection rates (2006–2016) and forecast of leprosy detection rates (2017–2020), Imperatriz, MA, Brazil (2006–2016).** (A) Observed data; (B) Adjusted data; (C) Estimated trend; (D) Expected trend; (E) Data forecasts; (F) 95% Confidence intervals.

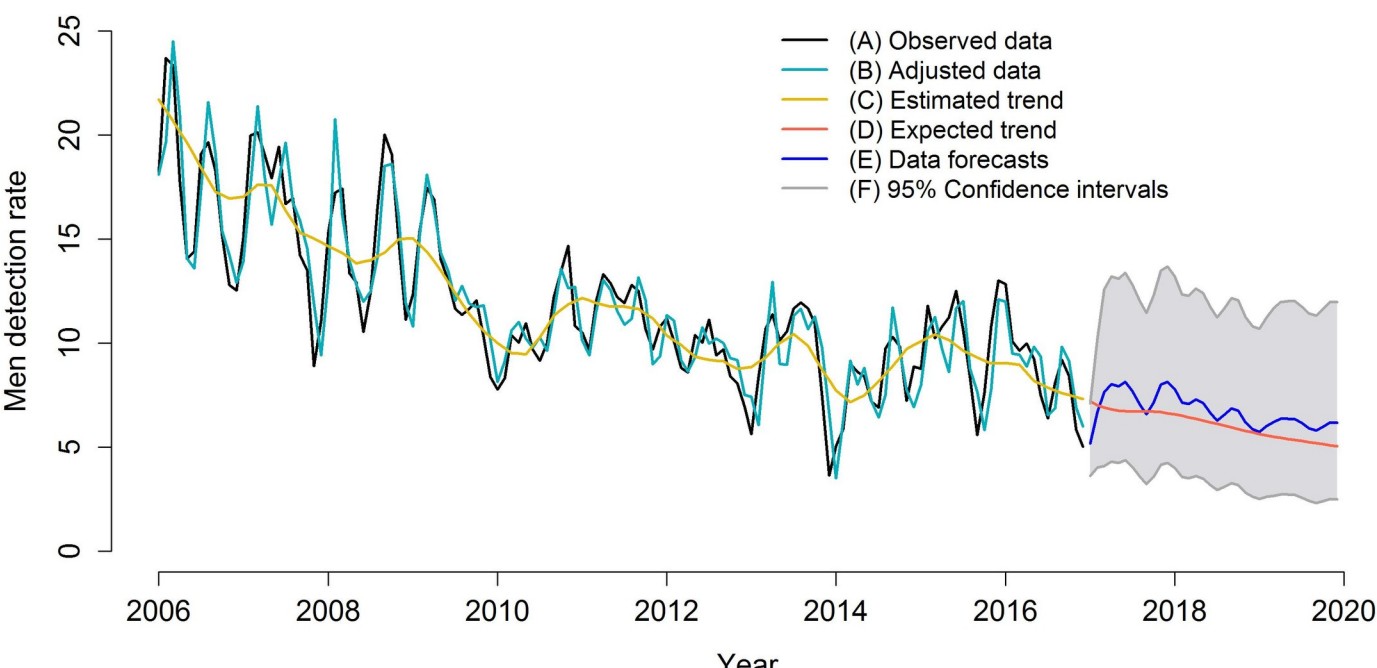

**Fig 6. SARIMA model (9,1,0) (1,0,0)$_{12}$ adjusted for leprosy detection rates among men (2006–2016) and forecast of leprosy detection rates among men (2017–2020), Imperatriz, MA, Brazil (2006–2016).** (A) Observed data; (B) Adjusted data; (C) Estimated trend; (D) Expected trend; (E) Data forecasts; (F) 95% Confidence intervals.

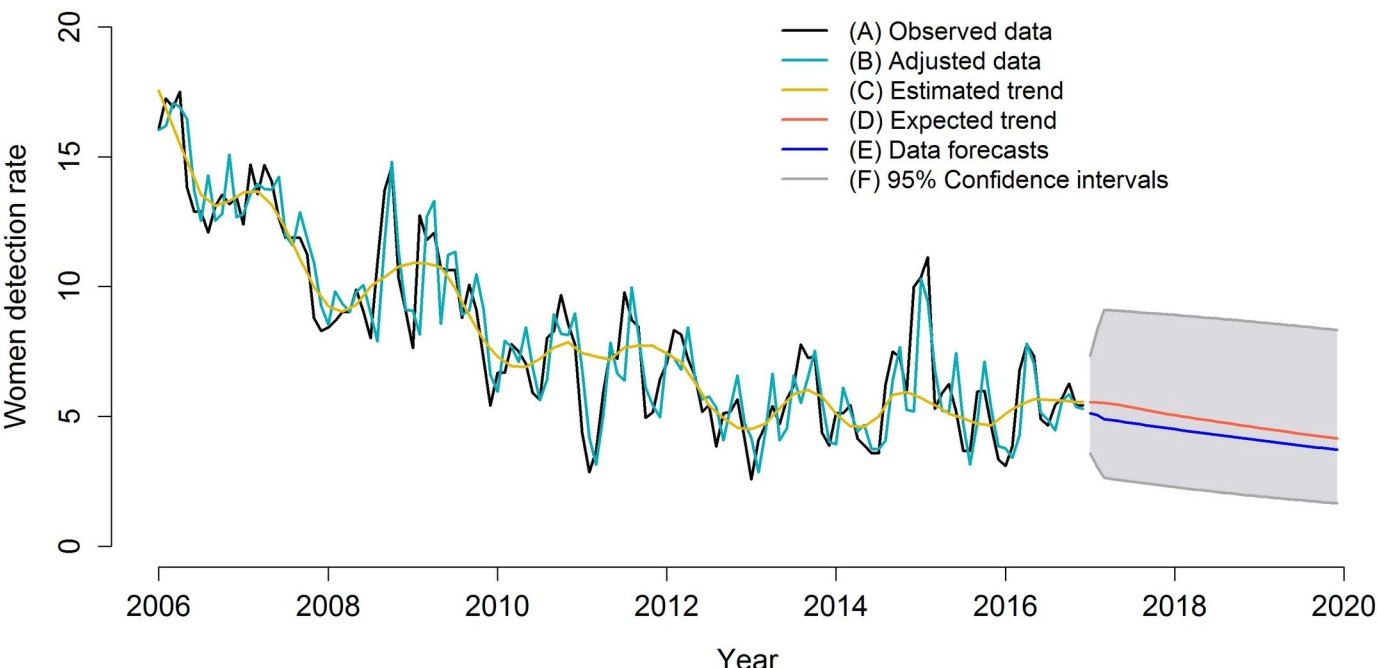

**Fig 7. ARIMA model (0,1,3) adjusted for leprosy detection rates among women (2006–2016) and forecast of leprosy detection rates among women (2017–2020), Imperatriz, MA, Brazil (2006–2016).** (A) Observed data; (B) Adjusted data; (C) Estimated trend; (D) Expected trend; (E) Data forecasts; (F) 95% Confidence intervals.

hyperendemic. Considering the demographic and social characteristics of the study context, the city is presenting rapid demographic growth, attracting immigrants from the north and northeast of the country due to its local economy, which may be influencing the hyperendemicity of the region under study [25, 26].

In this context, largest proportion of cases were male, which is in agreement with data provided in the literature, in which those affected by the disease in most world regions are predominately male, including in Brazil [27–29]. The time series of the ratio between the detection rates of men and women showed, over time, the detection rates in men are higher than that of women, reaching, at the end of the study period, a ratio of about 4: 1. In a few periods of the time series, women had slightly higher detection rates than men.

According to the literature, the occurrence of leprosy is higher in men, with a ratio of 2: 1 compared to women, also presenting a high risk of transmission [30, 31]. This greater occurrence of leprosy among men may be related to biological aspects, such as the role of the hormone testosterone, which may be involved in creating an environment favorable to the growth of *Mycobacterium leprae* causing a greater burden of the disease and the appearance of severe forms in the male's population [32–34].

Other factors may also explain the higher occurrence of the disease in men, as individual determinants, such as not seeking medical assistance or only later seeking health services, when compared to the practices of women. There are also operational issues concerning difficulties accessing health services in a timely manner due to an incompatibility between the men's and the units' working hours, a lack of a health policies directed to men, and restricted access to health information [15, 16].

The higher detection rates in men compared to women, in the research scenario, especially in the period from 2011 to 2016, are strong indications that transmission is higher in men. Such a finding may be an indication that the male sex is responsible for a large part of the

hyperendemicity of the study region, reinforcing the need to recognize and value men's health on the part of managers and health providers in the control of leprosy.

The discussion on gender issues allows the strengthening of professional health care practices aimed at men and women, aiming to achieve greater equity in public policies, especially in the context of neglected diseases, such as leprosy [10].

In terms of age, most of the individuals affected are aged between 30 and 59 years old, followed by the 15 to 29 year old group, though the population over 59 years old presents the highest detection rates.

Despite the high rates in the 30 to 59 year age group, it should be noted that the temporal trends in these intervals were decreasing in the study period. And the 15 to 29 year old interval also showed the highest average percentage decrease in the period from 2006 to 2016, which points to an indication of a reduction in bacillus transmission in the studied region [32, 35].

The group aged between 30 and 59 years old includes the Brazilian economically active population, interval that the disease hinders labor activities, forcing individuals to stop working or retire early, decreasing the quality of life of workers [28, 36, 37]. In this sense, health services should focus on preventive measures by actively seeking individuals in this age group, in addition to diagnosis of cases, providing timely treatment, and early identification of lesions, the purpose of which is also to prevent physical disability. Early interventions prevent or minimize the high social costs leprosy imposes from removing this population from productive activities and social relationships [28, 36–38].

According to the IBGE, in 2000, the Brazilian population aged over 60 years old amounted to 14.5 million people and this number currently surpasses 29 million and is expected to reach 73 million by 2060 [39]. In Imperatriz, in 2006, the estimated population aged 60 years old or over was 16,055 inhabitants, and in 2015 it increased to 21,0731 inhabitants, representing an increase of 31.25% in the period [40]. Considering the rapid aging process of the Brazilian and Imperatriz population, when leprosy is diagnosed and treated late, it leads to the functional loss of peripheral nerves and physical impairment, which, combined with the aging process and other comorbidities, contribute to elderly individuals' greater vulnerability and loss of autonomy [41].

Countries that registered a decrease in leprosy transmission, with subsequent elimination, observed a change in the profile of the disease, with a drop in detection in younger age groups and an increase in detection of elderly people [35, 42]. In Imperatriz, the high detection rates in the population aged 60 years old or over may be an indicative of a change in the epidemiological profile of the disease, despite the municipality still showing levels of hyperendemicity.

A decreasing tendency was found in the total detection rates of both men and women from 2006 to 2016, however, considering the age groups, women aged 60 years old or over and men aged less than 15 years old showed an increasing trend.

The downward trend in total detection, of men and women, possibly reflects the intensification of leprosy control actions in Maranhão in recent years. A study carried out in the municipalities of Maranhão (including Imperatriz) found that the decreasing trend in general detection are caused by actions to expand MDT, early detection of new cases, BCG vaccination of patient contacts, training of health professionals for diagnosis and active case-finding campaigns [43].

This study's findings show total and by-sex downward trends. More specifically, the figures reveal that the burden of the disease remains, considering that detection rates are systematically and persistently high for all the age groups addressed. Note the high occurrence of the disease among individuals below 15 years of age, with hyperendemicity parameters [9], confirming that the disease remains active in the community.

The occurrence of leprosy among individuals younger than 15 years old confirms that active foci of transmission remain with early exposure to *Mycobacterium leprae* [44]. Potential explanations include difficulty establishing a clinical diagnosis, disease-related stigma, and the weak health promotion and education process, needing improvements in leprosy control actions in these areas [15, 45, 46].

For example, studies conducted in two Brazilian regions in which leprosy is endemic an assessment of health services identified that the local primary health units did not present satisfactory performance in diagnosing individuals younger than 15 years old. The reason for this is that the services health diagnosis was conducted on request, however active case detection in the community is not being conducted. [46, 47].

Anchieta et al. (2019) [43] identified that Imperatriz showed a decreasing trend in the detection rate of children under 15 years old in the period from 2001 to 2015, a phenomenon explained by active case-finding national campaigns in the school-aged population in the years 2013 and 2016. Despite this decrease, the authors reaffirm that detection in children under 15 years old remains high in the municipality.

Brazil, since 2013, promotes the *"Campanha Nacional de Hanseníase, Verminoses Tracoma e Esquistossomose"* [National Campaign for Leprosy, Vermin, Trachoma and Schistosomiasis], which aims to identify cases of leprosy and provide timely treatment for the population that resides in municipalities in Brazil's endemic states, such as Maranhão. The campaign is aimed at students aged 5 to 14 years old, involving approximately 6 million students [48, 49].

Although the present study does not measure the impact of this action, the hypothesis arises that the National Campaign for Leprosy may be influencing the detection of cases under the age of 15 years old in the municipality, especially due to the rates found in the investigated period. It should be noted that the beginning of the campaign (2013) is concomitant with the beginning of the growth trend in men under 15 years old (2014).

Similar to other Brazilian studies, higher detection rates were found among male elderly individuals; however, the growing detection rates found among women aged 60 years old or over differ from other studies conducted in Brazil. According to Monteiro et al. (2013) [36], 60.3% of the leprosy patients located in the north of Brazil were male individuals aged over 60 years old, while Nobre et al. (2017) [32] determined that 15.11% more men than women were affected by the disease.

Studies conducted in India [50], China [51] and Colombia [52] report that women seek treatment later than men, a phenomenon that is mainly related to the stigma having a stronger affect for women than men in these countries, leading to late diagnoses and treatment. A literature review intended to identify the factors that prevent the early detection of leprosy among women shows that in some countries, the diagnosis and onset of the first symptoms take double the time among women, on average, compared to men, in addition to suggesting that women are more likely to initiate treatment late [53].

In Imperatriz, during the study period, no specific actions were found for women aged 60 years old or over that could explain the growing trend in this age group. The Enhanced Global Strategy for Further Reducing the Disease Burden Due to Leprosy (2011–2015) from WHO proposed the inclusion of female leprosy cases indicator among the total number of new cases, in order to assess and ensure that women are having adequate access to leprosy diagnostic services [54].

The implementation of this strategy and the creation of this indicator may have impacted the detection among women in the studied scenario, especially in women over 59 years old, considering that from 2011 onwards, the trend of this range shows an increasing behavior until 2016, concurrent with the period of validity of the strategy (2011–2015).

This study's findings lead to a discussion regarding the profile of the disease in the context of the studied scenario, revealing that the trends found in this study period indicate that males under 15 years old and women aged 60 years old or over showed an increasing trend in the detection rate from 2006 to 2016. These findings indicate that health services should direct efforts to detect cases of leprosy actively, considering that having individuals younger 15 years old affected by the disease indicates active transmission within a household and/or social group. The results also indicate the need to maintain actions to diagnose the disease earlier in the population aged 60 years old or over especially because this age group presents the highest detections rates in the investigated scenario. Future studies should be carried out in order to understand why women have an increasing trend in detection in the age group above 59 years.

In regard to the temporal modeling step, the SARIMA models selected for the total detection rates and rates according to sex presented adequate adjustments, providing efficacious models to capture the data's dependence structure; that is, the models effectively describe the variability of detection rates over time. Additionally, the models show downward trends for the three detection rates (total, men and women) in the predictive model for the 2017–2020 period.

The leprosy detection rates trends over the years in both the Brazilian and international contexts have depended considerably on operational factors, especially before 2000, due to the intensification of active search for cases to meet the elimination goal proposed by the WHO. Starting in 2001, a decline in the detection rates of new cases was observed, and since 2005 a stability of this indicator, caused mainly by a decrease in the intensive search for cases in many countries [55, 56]. The decreasing and stationary trends may indicate unchanged operational circumstances, indicating that transmission by *Mycobacterium leprae* is in progress [55].

Another aspect possibly related to the stability and decrease of leprosy concerns the estimated large number of non-detected cases. More than four million undetected cases are estimated from 2000 to 2020 worldwide, which implies a large number of people will remain undiagnosed and untreated [56].

In Norway, where leprosy was a serious public health problem in the 19th century, the reduction in transmission was accompanied by a change in the epidemiological profile of the disease, with a decrease in cases in young age groups and an increase in the proportion of elderly people among the new cases [35]. In our study, trends and forecasts of decreasing of total and by sex detection, accompanied by high detection rates in the age groups of 60 years old or older, may be indicative that leprosy transmission is decreasing. Despite this possible scenario of decreased leprosy and changes in the profile of patients, the state of Maranhão and the city of Imperatriz are hyperendemic for the disease.

A study conducted in India, Brazil and Indonesia identified that the incidence of leprosy up to 2020 will decrease and meet the elimination goal at a national level, though its elimination will not be possible for the highly endemic regions in these countries [57]. According to the authors, leprosy will likely remain a problem in endemic regions (states, districts, provinces, cities, with large populations), accounting for most of a country's cases.

The national forecasts of leprosy detection rates may provide a biased view of the disease situation, considering that these rates are masked by the large population size of each country [57]. Focusing on the regions of a country with high endemicity, such as the one addressed in this study, will give a more realistic representation of the current situation of a country, more accurately reflecting that the distribution of leprosy is becoming increasingly localized [57].

The results of the adjusted models and trends show a decrease in the total detection rates, as well as in detection rates of men and women separately in the study period. However, considering the forecasts and trends, leprosy will remain endemic and the WHO global goals to

decrease the disease's burden and eliminate the transmission of leprosy by 2020 may not be met.

This study's limitations include the fact a secondary database was used, with inconsistent quality and quantity of information, with the potential presence of ignored or incomplete data.

In conclusion, the results show downward trends of detection rates, in total and by sex. Despite decreasing trends, growing trends were found in terms of age; men aged below 15 years old and women aged 60 years old or over presented increasing detection rates, which is relevant in terms of public policies and strategic actions.

The models and forecasts for total detection rates, as well as detection rates for men and women, revealed downward trends in the study period. Leprosy, however, remains very frequent with hyperendemicity levels, making it difficult to decrease the disease's burden and eliminate of transmission by 2020.

## Supporting information

**S1 File. STROBE statement—checklist of items that should be included in reports of observational studies.**
(PDF)

**S1 Dataset. Minimal anonymized data set.**
(XLSX)

## Acknowledgments

The authors would like to thank the Health Surveillance Service of the Imperatriz Regional Health Management Unit, of the state government of Maranhão for making the data available.

## Author Contributions

**Conceptualization:** Antônio Carlos Vieira Ramos, Dulce Gomes, Marcelino Santos Neto, Thaís Zamboni Berra, Ricardo Alexandre Arcêncio.

**Formal analysis:** Antônio Carlos Vieira Ramos, Dulce Gomes, Marcelino Santos Neto, Thaís Zamboni Berra.

**Investigation:** Antônio Carlos Vieira Ramos, Dulce Gomes, Marcelino Santos Neto, Thaís Zamboni Berra, Ivaneliza Simionato de Assis, Mellina Yamamura, Juliane de Almeida Crispim, José Francisco Martoreli Júnior, Alexandre Tadashi Inomata Bruce, Felipe Lima dos Santos, Ludmilla Leidianne Limirio Souza, Yan Mathias Alves, Hamilton Leandro Pinto de Andrade, Marcos Augusto Moraes Arcoverde, Flávia Meneguetti Pieri, Ricardo Alexandre Arcêncio.

**Methodology:** Antônio Carlos Vieira Ramos, Dulce Gomes, Marcelino Santos Neto, Thaís Zamboni Berra.

**Project administration:** Antônio Carlos Vieira Ramos, Ricardo Alexandre Arcêncio.

**Supervision:** Ricardo Alexandre Arcêncio.

**Writing – original draft:** Antônio Carlos Vieira Ramos, Dulce Gomes, Marcelino Santos Neto, Thaís Zamboni Berra, Ivaneliza Simionato de Assis, Mellina Yamamura, Juliane de Almeida Crispim, José Francisco Martoreli Júnior, Alexandre Tadashi Inomata Bruce, Felipe Lima dos Santos, Ludmilla Leidianne Limirio Souza, Yan Mathias Alves, Hamilton

Leandro Pinto de Andrade, Marcos Augusto Moraes Arcoverde, Flávia Meneguetti Pieri, Ricardo Alexandre Arcêncio.

**Writing – review & editing:** Antônio Carlos Vieira Ramos, Mellina Yamamura, Juliane de Almeida Crispim, José Francisco Martoreli Júnior, Flávia Meneguetti Pieri, Ricardo Alexandre Arcêncio.

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
