## [Decision Letter · Decision Letter 0]

13 Mar 2020

PONE-D-20-00967

Trends and forecasts of leprosy for a hyperendemic city from Brazil’s northeast: Evidence from an eleven-year time-series analysis

PLOS ONE

Dear Mr Ramos,

Thank you for submitting your manuscript to PLOS ONE. After careful consideration, we feel that it has merit but does not fully meet PLOS ONE’s publication criteria as it currently stands. Therefore, we invite you to submit a revised version of the manuscript that addresses the points raised during the review process.

Please respond to all reviewer feedback.

We would appreciate receiving your revised manuscript by Apr 27 2020 11:59PM. To enhance the reproducibility of your results, we recommend that if applicable you deposit your laboratory protocols in protocols.io, where a protocol can be assigned its own identifier (DOI) such that it can be cited independently in the future. For instructions see: http://journals.plos.org/plosone/s/submission-guidelines#loc-laboratory-protocols

We look forward to receiving your revised manuscript.

Kind regards,

Eyal Oren, Ph.D.

Academic Editor

PLOS ONE

Journal Requirements:

4. We note that Figure 1 in your submission contains map images which may be copyrighted. All PLOS content is published under the Creative Commons Attribution License (CC BY 4.0), which means that the manuscript, images, and Supporting Information files will be freely available online, and any third party is permitted to access, download, copy, distribute, and use these materials in any way, even commercially, with proper attribution. For these reasons, we cannot publish previously copyrighted maps or satellite images created using proprietary data, such as Google software (Google Maps, Street View, and Earth). For more information, see our copyright guidelines: http://journals.plos.org/plosone/s/licenses-and-copyright.

Reviewers' comments:

Reviewer's Responses to Questions

**Comments to the Author**

1. Is the manuscript technically sound, and do the data support the conclusions?

Reviewer #1: Yes

Reviewer #2: Yes

2. Has the statistical analysis been performed appropriately and rigorously? 

Reviewer #1: Yes

Reviewer #2: Yes

3. Have the authors made all data underlying the findings in their manuscript fully available?

Reviewer #1: Yes

Reviewer #2: No

4. Is the manuscript presented in an intelligible fashion and written in standard English?

Reviewer #1: Yes

Reviewer #2: Yes

5. Review Comments to the Author

Reviewer #1: 1. Summary of the research In your own words, summarize the main research question, claims, and conclusions of the study. Provide context for how this research fits within the existing literature. Discuss the manuscript’s strengths and weaknesses and your overall recommendation.

The objective of this research was to estimate the temporal trends of leprosy according to sex and age groups, and to estimate and predict the progression of leprosy in Imperatriz, Maranhão, Brazil, a hyperendemic city in the northeast of Brazil.

Through the authors’ research, they found that while new case detection rates are declining in total, that data stratified by age and sex shows a different trajectory in certain groups. Namely that case detection rates among those less than 15 years of age and among women over 60 show an increase. Additionally the data show that those in the age range of 30 to 59 are the largest proportion of those affected by leprosy, even with the downward trend in case detection.

The discussion of results and conclusions are sound and recommend continued active surveillance to reduce leprosy transmission in highly endemic regions, even though overall new case detection is declining in the country. Additionally, discussion of social issues of men and women in relation to accessing health services and stigma is well documented in the manuscript and cites literature to support the recommendations.

The discussion of the final models for leprosy case detection show a decrease in total and among men and women separately. While I am not familiar with the epidemiology in predictive modeling utilized, the results are well described and match the findings.

Authors discuss limitations of using a secondary dataset. All ethics considerations have been addressed sufficiently.

Overall I found this manuscript to be very comprehensive. The research is important, which the authors state, because new case incidence rates are remaining stable or are not declining at the rate seen in the rest of the country, indicating continued active transmission. Additionally the prediction is that high case detection rates will continue to be a burden on the population in this hyperendemic area of Brazil. The study meets the criteria for publishing and I recommend publishing after minor edits.

2. Examples and evidence Major issues Major issues must be addressed in order for the manuscript to proceed. Focus on what is essential for the current study, not the next step in the research. Put these items in a list and be as specific as possible. Minor issues Mention additional things the authors should do to improve the manuscript. Typically these will be changes that would not affect the overall conclusions.

I found no major issues to be addressed. A recommendation that might strengthen the paper is to include significant tests for discussion of case detection in men in comparison to women. Since men have a higher case detection, it might be interesting for the reader to know if this is significantly higher than case detection in women.

The literature cited provides good examples of why men may have more difficulty in accessing health services. However, the “greater occurrence” of leprosy is related to transmission and not in accessing services. This may be a factor in delayed diagnosis and ongoing transmission as authors clearly state, but it might be interesting to include literature on transmission risk in men. Alternatively, since the data show trends, revising the language to exclude reasons for occurrence in exchange for risks for delayed diagnosis may be more appropriate.

In line 395 “The decrease in the rate drop” may be confusing. I would suggest to reword this so that the reader understands that the rate drop is not decreasing at the same rate as the time period prior to the year 2000. In line 454 there is a similar statement that “a rapid and continuous decline was observed, caused mainly by a decrease in the intensive search for cases in many countries”. Is this referring to a stability in the number of cases, indicating maintenance of leprosy during this time period? You may consider rewording.

There is a repeated phrase in lines 433 and 435.

Recommendations from the authors that delay in diagnosis will increase clinical severity of the disease and impairment and also contributes to on-going transmission shows the need for continued active leprosy surveillance to reach elimination goals locally in hyperendemic areas.

After minor edits, I recommend to publish this research.

3. Other points (optional) If applicable, add confidential comments for the editors. Raise any concerns about the manuscript that the editors may need to consider further, such as concerns about ethics. Do not use this section for your overall critique. Also mention whether you might be available to look at a revised version.

No other points to address.

Reviewer #2: Data from the Brazilian National Leprosy Database usually are available to the public through the Internet. Please comment on which data in your study are not available.

Specific comments

Abstract

Lines 40-41 – please remove the conjunction “while” from line 40 or the conjunction “and” from line 41

Lines 43-44 – Table 1 shows positives AMPC for rates among men aged less than 15 years old and among women over 60 years old, but not among women younger than 15 yo.

Introduction

Lines 59-60 – please use the term “elimination as a public health problem” and include the term “prevalence” before the criteria “<1 case every 10,000 inhabitants”. This is different from elimination of transmission. Here is a good paragraph in your paper to make this difference clear to the reader.

Line 64 – reference 4 is unnecessary once it is pointed at the end of the same paragraph

Line 66 – replace the term “degree” by “grade 2” or “visible deformities”

Line 69 – use the term “global detection” instead of “total”

Line 71 – use the term “total” before “cases reported”

Line 73 – the reference 6 show a link related to leprosy in Maranhao, why is it used for these data in the text?

(http://tabnet.datasus.gov.br/cgi/dhdat.exe?hanseniase/hantfma17.def )

Lines 59-60 – please provide a reference for the following statement about high rates of the disease among Brazilian regions.

Line 77 – please remove the verb “was”

Lines 82-84 – we suggest to add that the data on G2D is the “rate per 100,000 inhabitants” to make sure the reader will not misunderstand this information as the percentage of G2D among new cases.

Line 91 – reference 8 is unnecessary once it is pointed at the end of the same paragraph

Line 96 – we suggest to include “the disease” before the verb presents

Lines 97-98 – Please provide a reference to the statement: “that is, men are more frequently affected than women in most regions of the world (including Brazil)”

Line 104 – Please clarify where Souza’s study was carried out.

Line 119 – The period (2006-2016) should not be part of the figure title, unless the municipality area has changed

Lines 126-130 – please specify the year for the count or estimation of that population and for main social indicators

Lines 142-143 – Please rephrase the following: “The variables adopted in this study include date when leprosy was reported, age and sex” once “age and sex” are unchangeable individual characteristics differently from “date” and clarify if you considered the date of diagnosis or the date when the case was reported to the system.

Lines 153-155 – Please specify how total and specific intercensal population was calculated.

Results

Lines 223-231 – Table 1 shows a continuous increase for new cases detection rates along lifetime, an interesting data not cited by authors.

We suggest that table 1 shows the total number of cases, and its related rates and AMPC.

Lines 242-243 – The NCDR decreasing trend looks different between the first and second 5 years period, showing a stable tendency especially from 2013 to 2016 which may be further investigated and discussed.

Lines 249-254 – we suggest to say that figure 3 shows a decreasing trend for all age-groups and sex, except for men younger than 15 and for women older than 59 yo, which reflects specific AMPC by age and sex. For women younger than 15 and aged 15 to 29 yo there was a slight increase in the last year.

This is different from saying that there is an “increasing tendency” for women younger than 15 yo. The same slight increase in 2016 was registered for women aged 15-29 yo but was not mentioned by authors.

It is important to observe that the older group is not “over 60” but “aged 60 or more”.

Why figure 2 was titled “Leprosy Trend” while figure 3 was titled “Estimated Leprosy Trend”?

There is little additional information on figure 4 comparing to figure 3. We suggest excluding this figure. The text from line 265 to 277 may be maintained as comments related to figure 3.

Discussion

Lines 344-346 – First paragraph just repeat the study’s objective which is unnecessary.

Lines 347-353 – Main results of this study was a downward trend of leprosy in the last decade, which will be maintained in the next years accordingly to author’s predictions. Thus, apparently this is not compatible with an “increasing circulation of leprosy bacillus in the region”.

Demographic growth and immigration may be linked to the very high rates of leprosy observed in Imperatriz, but the text should be better explained separating that from the apparent decline, which also needs to be discussed.

Lines 359-367 – Authors comment only about operational factors possibly related to higher rates of leprosy among men, but other hypotheses as biological aspects are discussed on literature and should be considered. This is pointed by some authors already cited as references for your study.

Lines 369 – The study subgroup was not over 60 years old, but over 59.

Lines 370-378 – It is important to observe not only that this age group (30-59 yo) presented the highest detection rates, but also to highlight that sharper decreases were registered for them in both sexes (table 1), which may reflects reduction of transmission.

Lines 379-385 – As the study was carried out in one municipality only, data from Brazilian population are not adequate to discuss your results. We suggest reporting on Imperatriz population composition and about any changes observed during the study period, like a population aging or decrease on birthrate.

Lines 367-389 – An increasing in AMPC is clear for NCDR in female aged 60 or more and for young men, but the same is not so clear for young women, despite a slight late increase in the last year, which cannot be considered as a tendency.

This needs to be discussed? What could explain this decreases and increases? MDT? Case search activities for children like school campaign? any specific campaign for the elderly? Do women respond better to those activities?

Lines 396-406 – Recent leprosy campaigns for school children in Brazil became globally know, they did not happen in your study area?

Would you consider a possible decline in leprosy transmission and not only a decline from operational factors? If this is not a possible hypothesis, please make some comments.

Lines 415-418 – Please provide some data from your study that could support the idea of underreporting of cases in children. Did they present a high grade 2 disability rate, for example? Were most of your cases classified as multibacillary despite paucibacillary expected predominance among children? Is the coverage of household contact examination very low?

Lines 419-430 – The important progressive increase in NCDRs for elderly females observed in your study from 2011 is really an intriguing result. This was the only rate to reach the same level from the beginning of your time series.

Your discussion addressed issues related to a possible later diagnosis for women than men, but you did not present any data from Imperatriz to support this hypothesis, for example, a recent change with increase of grade 2 disability rate for this sex and age group would be an indicator for that. We suggest a deeper investigation on possible specific actions taken for this sex and age group that could have increased leprosy detection from 2011.

Lines 437-440 – The text was repeated, please correct it.

In your whole study period NCDRs for children younger than 15 yo ranged roughly from 1 to 10 new cases per 100,000 children. On the other hand, you reported that NCDRs for the elderly fluctuated between 10 to 60 new cases per 100,000 people aged 60 or more. Why do you affirm that “individuals younger than 15 years old are more vulnerable to the disease”?

Lines 440-442 – You did not report any data about physical disability among the elderly female.

Line 446 – please replace “early” by “earlier”.

Lines 458-462 – Some published papers registered a real decrease of leprosy in some countries, such as Norway, where there were a decrease of transmission followed by a change in epidemiological pattern of disease towards older ages. This hypothesis should also be considered and discussed by authors in addition to their operational explanations for decreased chances of diagnosis.

Lines 473-478 – We suggest a better definition of “elimination of leprosy as a public health problem” and of “elimination of transmission of leprosy”, which can be explained in the introduction of this paper. This will avoid misunderstanding of readers, especially at the end of the discussion section.

Line 491 – we suggest reviewing your affirmation that there was a growing trend for women aged less than 15, despite the slight increase in 2016.

6. PLOS authors have the option to publish the peer review history of their article (what does this mean?). If published, this will include your full peer review and any attached files.

Reviewer #1: No

Reviewer #2: No

---

## [Author Response · Author response to Decision Letter 0]

22 Apr 2020

Dear Editor, 

Many thanks for your reply and your reviewers' comments about our manuscript “PONE-D-20-00967 Trends and forecasts of leprosy for a hyperendemic city from Brazil’s northeast: Evidence from an eleven-year time-series analysis”.

The comments were appropriate to qualify/ improve the manuscript. We have forwarded a letter with the changes made in the manuscript, presenting our response for each comment from reviewers. Many thanks for your comments.

In the version presented in the journal, we adapted the question of data availability, they were collected upon request and approval by an ethics committee, but for the analyzes we did not use any variable to identify the cases. Since the data has no restrictions, we uploaded the minimum set of anonymous data to replicate the study's findings.

In relation to Fig 1, it was constructed using data available from a Brazilian public repository, the Brazilian Institute of Geography and Statistics (IBGE) (https://www.ibge.gov.br/estatisticas/downloads-estatisticas.html), open access, without restrictions and without copyright. The data used for making the map (Fig 1) refer to files in the Shapefile extension, which were manipulated through the ArcGis® 10.6 software for making the maps, with the authors being entirely responsible for drawing the figure. The authors made this issue of copyright clear in the submission.

Below, we respond point by point to the comments of Reviewer #1 and Reviewer #2.

Kind regards,

Ramos et al.

Reviewers' comments:

Reviewer #1: 

I found no major issues to be addressed. A recommendation that might strengthen the paper is to include significant tests for discussion of case detection in men in comparison to women. Since men have a higher case detection, it might be interesting for the reader to know if this is significantly higher than case detection in women.

Authors: A test was included that verified the time series of the relationship between the detection rate of men and women (Fig 3), showing that over time, the rate of men is higher than that of women:

Lines 267 – 276: “The time series of the ratio between the detection rates of men and women is shown in Fig 3. The red line indicates the situation in which both rates would be equal (numerator equal to the denominator), and the blue line indicates the time period in that the ratio shows a change in behavior. Over time, the rate of men is generally higher than that of women, noting that, in the period from January 2006 to approximately September 2010, the rate of men reached, at most, double that of women; in January 2011 this difference exceeded the triple, and the quadruple in 2016.

Women showed slightly higher rates than men between September and December 2006, September 2013, between October 2014 and January 2015 and, finally, in July 2015.”

The literature cited provides good examples of why men may have more difficulty in accessing health services. However, the “greater occurrence” of leprosy is related to transmission and not in accessing services. This may be a factor in delayed diagnosis and ongoing transmission as authors clearly state, but it might be interesting to include literature on transmission risk in men. Alternatively, since the data show trends, revising the language to exclude reasons for occurrence in exchange for risks for delayed diagnosis may be more appropriate.

Authors: Evidence from the literature on the biological aspects of leprosy and on the risk of transmission of the disease in men, which affects men more strongly, was included. The authors continued to discuss the difficulties in accessing health services, since the set of biological aspects and whether health services are the main determinants for the greater involvement of men by leprosy, in Brazil and in the study scenario.

The excerpt referring to this discussion was adequate to facilitate the reading:

Lines 375 – 395: “According to the literature, the occurrence of leprosy is higher in men, with a ratio of 2: 1 compared to women, also presenting a high risk of transmission [30,31]. This greater occurrence of leprosy among men may be related to biological aspects, such as the role of the hormone testosterone, which may be involved in creating an environment favorable to the growth of Mycobaterium leprae causing a greater burden of the disease and the appearance of severe forms in the male’s population [32,33,34].

Other factors may also explain the higher occurrence of the disease in men, as individual determinants, such as not seeking medical assistance or only later seeking health services, when compared to the practices of women, and also related to operational issues concerning difficulties accessing health services in a timely manner due to an incompatibility between the men’s and the units’ working hours, a lack of a health policies directed to men, and restricted access to health information [15,16]. 

The higher detection rates in men compared to women, in the research scenario, especially in the period from 2011 to 2016, are strong indications that transmission is higher in men, occurring continuously and increasing over time. Such a finding may be an indication that the male sex is responsible for a large part of the hyperendemicity of the study region, reinforcing the need to recognize and value men's health on the part of managers and health providers in the control of leprosy. 

The discussion on gender issues allows the strengthening of professional health care practices aimed at men and women, aiming to achieve greater equity in public policies, especially in the context of neglected diseases, such as leprosy [10].”

In line 395 “The decrease in the rate drop” may be confusing. I would suggest to reword this so that the reader understands that the rate drop is not decreasing at the same rate as the time period prior to the year 2000. 

Authors: The sentence that the reviewer indicated for correction was removed from the manuscript after further corrections.

In line 454 there is a similar statement that “a rapid and continuous decline was observed, caused mainly by a decrease in the intensive search for cases in many countries”. Is this referring to a stability in the number of cases, indicating maintenance of leprosy during this time period? You may consider rewording.

Authors: Since 2001, in the international context, there has been a decline in the rates of detection of new cases, and since 2005 a stability of this indicator, caused by a decrease in the intensive search for cases. This stability of the indicator points to a maintenance of the disease, in which transmission is ongoing, due to this stability.

The excerpt was revised according to the reviewer's suggestion for easier reading.

Lines 514-518: “Starting in 2001, a decline in the detection rates of new cases was observed, and since 2005 a stability of this indicator, caused mainly by a decrease in the intensive search for cases in many countries [55,56]. The decreasing and stationary trends may indicate unchanged operational circumstances, indicating that transmission by Mycobacterium leprae is in progress [55].”

There is a repeated phrase in lines 433 and 435.

Authors: The sentences were corrected.

Reviewer #2: 

Data from the Brazilian National Leprosy Database usually are available to the public through the Internet. Please comment on which data in your study are not available.

Authors: Among the data available from the database we use are the home addresses of leprosy cases, which were collected through the health department of the state of Maranhão, after approval by the research ethics committee. The use of residential addresses has the purpose of georeferencing the cases, however this method was not used in this work. As the data in this article does not involve variables to identify cases, we will provide a minimum set of data, meeting the requirements of the journal PLOS ONE.

Abstract

Lines 40-41 – please remove the conjunction “while” from line 40 or the conjunction “and” from line 41

Authors: The “while” and “and” conjunctions have been removed.

Lines 43-44 – Table 1 shows positives AMPC for rates among men aged less than 15 years old and among women over 60 years old, but not among women younger than 15 yo.

Authors: The sentence was corrected, women under 15 years of age did not have a positive AMPC:

Lines 42-44: “Detection rates in total and by sex presented a downward trend, though rates stratified according to sex and age presented a growing trend among men aged less than 15 years old and among women aged 60 years old or over.”

Introduction

Lines 59-60 – please use the term “elimination as a public health problem” and include the term “prevalence” before the criteria “<1 case every 10,000 inhabitants”. This is different from elimination of transmission. Here is a good paragraph in your paper to make this difference clear to the reader.

Authors: The term “elimination as a public health problem” and “prevalence” were inserted, according to the suggestion.

In the paragraph it was presented about the criterion of elimination of the disease (prevalence <1 case every 10,000 inhabitants) and an explanation about the goals of elimination of transmission (zero transmission) that are great challenges for a world without leprosy:

Lines 59 – 66: “Even though leprosy has been eliminated as public health problem in many countries in the world (prevalence <1 case every 10,000 inhabitants) since the year 2000, leprosy still persists in developing countries as a serious public health problem [2,3]. After the introduction of Multidrug Therapy (MDT) and the high vaccination coverage of the Bacillus Calmette-Guérin (BCG), especially in children, a burden of leprosy has decreased considerably worldwide. However, in some nations the elimination of the disease (zero transmission) and decreased detection of new cases continue to be important challenges for a world without leprosy [4].”

Line 64 – reference 4 is unnecessary once it is pointed at the end of the same paragraph

Authors: The text has been revised and the reference is only at the end of the paragraph.

Line 66 – replace the term “degree” by “grade 2” or “visible deformities”

Authors: The term “degree” has been replaced by “grade 2” (Line 70).

Line 69 – use the term “global detection” instead of “total”

Authors: The term “global detection” was used according to the reviewer's suggestion:

Lines 73 – 75: “The global detection rate for leprosy in 2018 was 1.93 cases/100,000 inhabitants and the countries that presented the highest rates were India, Brazil and Indonesia, responsible for 79.6% of the cases reported [5].”

Line 71 – use the term “total” before “cases reported”

Authors: The term was included according to the suggestion.

Line 73 – the reference 6 show a link related to leprosy in Maranhao, why is it used for these data in the text? (http://tabnet.datasus.gov.br/cgi/dhdat.exe?hanseniase/hantfma17.def )

Authors: The link was corrected. It should refer to epidemiological data on leprosy in Brazil. The authors corrected this error in the references:

Brasil. Ministério da Saúde. Departamento de Informática do SUS (DATASUS). Acompanhamento da Hanseníase - BRASIL (2001-2017) [Internet]. 2020 [cited 08 Apr 2020]. Available from: http://tabnet.datasus.gov.br/cgi/tabcgi.exe?sinannet/hanseniase/cnv/hanswuf.def

Lines 59-60 – please provide a reference for the following statement about high rates of the disease among Brazilian regions.

Authors: A reference related to the passage in question was inserted:

7. Ribeiro MDA, Silva JCA, Oliveira SB. Estudo epidemiológico da hanseníase no Brasil: reflexão sobre as metas de eliminação. Rev Panam Salud Publica. 2018;42:e42. https://doi.org/10.26633/RPSP.2018.42

Line 77 – please remove the verb “was”

Authors: Removed the verb “was”.

Lines 82-84 – we suggest to add that the data on G2D is the “rate per 100,000 inhabitants” to make sure the reader will not misunderstand this information as the percentage of G2D among new cases.

Authors: It has been added that the rate of GD2 is for 100,000 inhabitants:

Lines 86-88: “However, the rate of new cases with grade 2 physical disability per 100,000 inhabitants remained stable over the period, suggesting late diagnosis and possibly underreporting of cases.”

Line 91 – reference 8 is unnecessary once it is pointed at the end of the same paragraph

Authors: The reference has been removed and is only at the end of the paragraph.

Line 96 – we suggest to include “the disease” before the verb presents

Authors: Inserted (Line 100).

Lines 97-98 – Please provide a reference to the statement: “that is, men are more frequently affected than women in most regions of the world (including Brazil)”

Authors: A reference to the statement has been inserted, and the paragraph has been rewritten so that readers have a better understanding:

Lines 99 – 107: “In addition to the fact that leprosy is a tropical disease that has been neglected, its association with poverty and social inequality, the disease presents a sex-specific distribution in terms of morbidity [10–12]; that is, men are more frequently affected than women in most regions of the world (including Brazil) [13]. However, in many countries, women are late diagnosed and have a higher proportion of degrees of physical disability, in addition to the fact that the stigma of the disease is greater in women [14]. Leprosy is a disease known for leading to different representations and effects between men and women, in different social contexts, and as a consequence, it accentuates gender inequalities from the Brazilian sociocultural point of view [15].”

13. Britton WJ, Lockwood DNJ. Leprosy. Lancet. 2004;363:1209–19. https://doi.org/10.1016/S0140-6736(04)15952-7 

Line 104 – Please clarify where Souza’s study was carried out.

Authors: The study by Souza et al (2018) was carried out in the state of Bahia, northeast region of Brazil:

Lines 109-114: “A study carried out in the state of Bahia (Northeast region of Brazil) by Souza et al. (2018) analyzed temporal trends in terms of sex and verified that the disease behaves differently according to sex: there is a tendency for decreased detection coefficients among women, but detection coefficients remain stable among men, though the results were not statistically significant [15].”

Line 119 – The period (2006-2016) should not be part of the figure title, unless the municipality area has changed

Authors: The study period has been removed from the figure:

Fig 1. Location map of the study setting, Imperatriz, Brazil. (A) Brazil. (B) State of Maranhão. (C) City of Imperatriz. 

Lines 126-130 – please specify the year for the count or estimation of that population and for main social indicators

Authors: The year of the population estimate and social indicators is 2010, the year of the last Brazilian Demographic Census:

Lines 129-136: “Imperatriz is located 626 km from the capital of Maranhão, São Luís, and is the second largest city in the state and the 23rd largest city in the Brazilian northeast. According to the Demographic Census of the Brazilian Institute of Geography and Statistics (IBGE), in 2010, Imperatriz has a population of 247,505 inhabitants, with a demographic density of 180.79 inhabitants/km2 with a territorial area of 1,368,988 km2 [18]. In the same period, the main social indicators are: an illiteracy rate of 9.7%, Human Development Index (HDI) of 0.73 and Gini Index of 0.46. In terms of basic sanitation, 23% of the city has a sewage system and 86% has a drinking water supply [18,19].”

Lines 142-143 – Please rephrase the following: “The variables adopted in this study include date when leprosy was reported, age and sex” once “age and sex” are unchangeable individual characteristics differently from “date” and clarify if you considered the date of diagnosis or the date when the case was reported to the system.

Authors: The excerpt has been rewritten by the authors, making it clear that the date variable refers to when the case was registered on SINAN:

Lines 148-151: “The variables adopted in this study include date when leprosy cases were reported in the SINAN (notification date), age and sex. Data were collected at the health surveillance service from the city’s regional management unit, state government of Maranhão in May 2018.”

Lines 153-155 – Please specify how total and specific intercensal population was calculated.

Authors: It was specified how the specific and total population were calculated:

Lines 162-164: “The size of the resident populations used as the denominator was based on the 2010 Census and the intercensal estimates (2006–2016) elaborated by the IBGE.”

Results

Lines 223-231 – Table 1 shows a continuous increase for new cases detection rates along lifetime, an interesting data not cited by authors. We suggest that table 1 shows the total number of cases, and its related rates and AMPC.

Authors: It was mentioned in the description of the results that there is a continuous growth of new cases throughout life, and this result was discussed in the discussion section:

Lines 245-247: “There is a continuous increase in the rates of case detection as the age groups increase, with the lowest rates being for children under 15 years old and the highest for those aged 60 years old or over.”

Table 1 shows the total number of cases according to age groups and their respective AMPC (Line 252).

Lines 242-243 – The NCDR decreasing trend looks different between the first and second 5 years period, showing a stable tendency especially from 2013 to 2016 which may be further investigated and discussed.

Authors: This observation raised by the reviewer was presented in the results:

259-261: “Analyzing the comparison between the three trends, it is possible to observe stability in the period from 2014 to 2016.”

Lines 249-254 – we suggest to say that figure 3 shows a decreasing trend for all age-groups and sex, except for men younger than 15 and for women older than 59 yo, which reflects specific AMPC by age and sex. For women younger than 15 and aged 15 to 29 yo there was a slight increase in the last year. This is different from saying that there is an “increasing tendency” for women younger than 15 yo. The same slight increase in 2016 was registered for women aged 15-29 yo but was not mentioned by authors. It is important to observe that the older group is not “over 60” but “aged 60 or more”. Why figure 2 was titled “Leprosy Trend” while figure 3 was titled “Estimated Leprosy Trend”? There is little additional information on figure 4 comparing to figure 3. We suggest excluding this figure. The text from line 265 to 277 may be maintained as comments related to figure 3.

Authors: The reviewer's suggestions related to the explanation in Fig 3 were considered by the authors, making it clear that all age groups showed a decreasing trend, except men under 15 and women aged 60 or over. We make it clear that the oldest group is 60 and over, and not over 60:

282-290: “Tendency toward leprosy according to sex and age (Fig 4) shows a decreasing trend for all age groups and sex, except for men younger than 15 years old and women aged 60 years old or over, reflecting specifically the AMPC by age and sex. Men under 15 years old showed decreasing trends from 2006 to 2014, after which they showed an increasing trend until the end of the study period. Women aged 60 years old or over showed a peak of detection between the years 2008 and 2009, with a decrease until 2011, and subsequently a continuous growth trend until the year 2016. 

For women younger than 15 years old and aged between 15 to 29 years old there was a slight increase in the last year.”

There was a change in the order of the figures, Fig 3 (from the first submission) became Fig 4. The title of Fig 2 and Fig 4 were standardized, as both estimated the trends of leprosy:

Fig 2. Trend of leprosy total detection rates and detection rates among men and women. Imperatriz, MA, Brazil (2006-2016). (A) Total detection rate; (B) Men’s detection rate; (C) Women’s detection rate.

Fig 4. Trends of leprosy according to sex and age groups. Imperatriz, MA, Brazil (2006-2016). (A) Men younger than 15 years old; (B) Men aged between 15 and 29 years old; (C) Men aged between 30 and 59 years old; (D) Men aged 60 years old or over; (E) Women younger than 15 years old; (F) Women aged between 15 and 29 years old; (G) Women aged between 30 and 59 years old; (H) Women aged 60 years old or over.

Fig 4 (from the original version submitted) was deleted as suggested by the reviewer

Discussion

Lines 344-346 – First paragraph just repeat the study’s objective which is unnecessary.

Authors: The first paragraph that presents the objective of the study has been removed.

Lines 347-353 – Main results of this study was a downward trend of leprosy in the last decade, which will be maintained in the next years accordingly to author’s predictions. Thus, apparently this is not compatible with an “increasing circulation of leprosy bacillus in the region”. Demographic growth and immigration may be linked to the very high rates of leprosy observed in Imperatriz, but the text should be better explained separating that from the apparent decline, which also needs to be discussed.

Authors: In the excerpt pointed by the reviewer, the authors present the profile of Imperatriz, indicating that the city is hyperendemic for leprosy. The terms “increasing circulation of leprosy bacillus in the region” have been removed, and the authors have made it clear that the increase in immigration may be influencing the high rates observed in the municipality. The paragraph was rewritten for a better understand of the readers:

Lines 363-368: “In 2016, Imperatriz presented a total detection rate of 62.23 cases/100,000 inhabitants, which, according to parameters provided by the Ministry of Health, classifies the city as hyperendemic. Considering the demographic and social characteristics of the study context, the city is presenting rapid demographic growth, attracting immigrants from the north and northeast of the country due to its local economy, which may be influencing the hyperendemicity of the region under study [25,26].”

The issue of declining rates and forecasts was discussed later in the manuscript.

Lines 359-367 – Authors comment only about operational factors possibly related to higher rates of leprosy among men, but other hypotheses as biological aspects are discussed on literature and should be considered. This is pointed by some authors already cited as references for your study.

Authors: According to the reviewer's suggestion, a discussion of the role of the biological aspects of leprosy in men was brought up:

Lines 369-380: “In this context, most of the reported cases were males, which is in agreement with data provided in the literature, in which those affected by the disease in most world regions are predominately male, including in Brazil [27–29]. The time series of the ratio between the detection rates of men and women showed, over time, the detection rates in men are higher than that of women, reaching, at the end of the study period, a ratio of about 4: 1. Punctually, women had slightly higher detection rates than men.

According to the literature, the occurrence of leprosy is higher in men, with a ratio of 2: 1 compared to women, also presenting a high risk of transmission [30,31]. This greater occurrence of leprosy among men may be related to biological aspects, such as the role of the hormone testosterone, which may be involved in creating an environment favorable to the growth of Mycobacterium leprae causing a greater burden of the disease and the appearance of severe forms in the male’s population [32,33,34].”

Lines 369 – The study subgroup was not over 60 years old, but over 59.

Authors: Corrected (Line 397).

Lines 370-378 – It is important to observe not only that this age group (30-59 yo) presented the highest detection rates, but also to highlight that sharper decreases were registered for them in both sexes (table 1), which may reflects reduction of transmission.

Authors: We made it clearer that the 30-59 age groups had high detection rates, as well as decreasing trends over the study period. We also make it clear that, as the age groups increase, detection rates are also increasing, with the lowest rates in children under 15 and the highest in people aged 60 years old or over, indicating a change in the epidemiological profile of the disease and possible transmission reduction:

Lines 396-403: “In terms of age, most of the individuals affected are aged between 30 and 59 years old, followed by the 15 to 29 years old group, though the population over 59 years old presents the highest detection rates. 

Despite the high rates in the 30 to 59 year age group, it should be noted that the temporal trends in these intervals were decreasing in the study period. And the 15 to 29 years old interval also showed the highest average percentage decrease in the period from 2006 to 2016, which points to an indication of a reduction in bacillus transmission in the studied region [32,35].”

 Lines 422-427: “Countries that registered a decrease in leprosy transmission observed a change in the profile of the disease, with a drop in detection in younger age groups and an increase in detection of elderly people [35,42]. In Imperatriz, the continuous increase in the rate of detection of new cases over the age groups, especially the highest rates in the elderly, may by an indicative of a decrease in transmission, despite the municipality still showing levels of hyperendemicity.”

Lines 379-385 – As the study was carried out in one municipality only, data from Brazilian population are not adequate to discuss your results. We suggest reporting on Imperatriz population composition and about any changes observed during the study period, like a population aging or decrease on birthrate.

Authors: Data from the population of Imperatriz aged 60 years old or over were collected from 2006 to 2015 (population estimates), in order to discuss the results of the work and present the growth of this population in the period (31.25%):

Lines 413-421: “According to the IBGE, in 2000, the Brazilian population aged over 60 years old amounted to 14.5 million people and this number currently surpasses 29 million and is expected to reach 73 million by 2060 [39]. In Imperatriz, in 2006, the estimated population aged 60 years old or over was 16,055 inhabitants, and in 2015 it increased to 21,0731 inhabitants, representing an increase of 31.25% in the period [40]. Considering the rapid aging process of the Brazilian and Imperatriz population’s, when leprosy is diagnosed and treated late, it leads to the functional loss of peripheral nerves and physical impairment, which, combined with the aging process and other comorbidities, contribute to elderly individuals’ greater vulnerability and loss of autonomy [41].”

Lines 367-389 – An increasing in AMPC is clear for NCDR in female aged 60 or more and for young men, but the same is not so clear for young women, despite a slight late increase in the last year, which cannot be considered as a tendency. This needs to be discussed? What could explain this decreases and increases? MDT? Case search activities for children like school campaign? any specific campaign for the elderly? Do women respond better to those activities?

Authors: The authors agree with the points raised by the reviewer. Indeed, women under the age of 15 did not present a positive AMPC and did not show an increasing trend during the study period (only an increase in the last year, but it cannot be considered an increase in the trend). This was corrected in the discussion.

Evidence was brought from the literature that active search actions, expansion of MTD, high coverage of BCG and school campaigns are influencing the behavior of leprosy detection rates in the investigated scenario, causing a decreasing trend in recent years:

Lines 428-437: “A decreasing tendency was found in the total detection rates of both men and women from 2006 to 2016. However, when the AMPC and tendencies according to sex and age are verified, increasing tendencies are found for men aged less than 15 years old and among women aged 60 years old or over. 

The downward trend in total detection, of men and women, possibly reflects the intensification of leprosy control actions in Maranhão in recent years. A study carried out in the municipalities of Maranhão (including Imperatriz) found that the decreasing trend in general detection are caused by actions to expand MDT, early detection of new cases, BCG vaccination of patient contacts, training of health professionals for diagnosis and active case-finding campaigns [43].”

Lines 396-406 – Recent leprosy campaigns for school children in Brazil became globally know, they did not happen in your study area? Would you consider a possible decline in leprosy transmission and not only a decline from operational factors? If this is not a possible hypothesis, please make some comments.

Authors: Leprosy campaigns aimed at school-age children were carried out in the study area (National Campaign for Leprosy, Vermin, Trachoma and Schistosomiasis), which may have influenced the growing trend of men under 15 years of age, since the increase trend and coinciding with the campaign period.

The authors agree with the reviewer and discussed the issue of declining leprosy transmission, discussing our results with studies carried out in other countries, especially in Norway:

Lines 453-469: “Anchieta et al. (2019) [43] identified that Imperatriz showed a decreasing trend in the detection rate of children under 15 years old in the period from 2001 to 2015, a phenomenon explained by active case-finding campaigns in the school-aged population in the years 2013 and 2016. Despite this decrease, the authors reaffirm that detection in children under 15 years old remains high in the municipality. 

Brazil, since 2013, promotes the “Campanha Nacional de Hanseníase, Verminoses Tracoma e Esquistossomose” [National Campaign for Leprosy, Vermin, Trachoma and Schistosomiasis], which aims to identify cases of leprosy and provide timely treatment for the population that resides in municipalities in Brazil's endemic states, such as Maranhão. The campaign is aimed at students aged 5 to 14 years old, involving approximately 6 million students [48,49]. 

Although the present study does not measure the impact of this action, the hypothesis arises that the National Campaign for Leprosy may be influencing the detection of cases under the age of 15 years old in the municipality, especially due to the rates found in the investigated period, as well as, it is coincidental that the beginning of the campaign (2013) is concomitant with the beginning of the growth trend in men under 15 years old (2014).”

Lines 523-531: “In Norway, where leprosy was a serious public health problem in the 19th century, the reduction in transmission was accompanied by a change in the epidemiological profile of the disease, with a decrease in cases in young age groups and an increase in the proportion of elderly people among the new cases [35]. In our study, trends and forecasts of decreasing of total and by sex detection, accompanied by high detection rates in the age groups of 60 years old or older, may be indicative that leprosy transmission is decreasing. Despite this possible scenario of decreased leprosy and changes in the profile of patients, the state of Maranhão and the city of Imperatriz are hyperendemic for the disease.”

Lines 415-418 – Please provide some data from your study that could support the idea of underreporting of cases in children. Did they present a high grade 2 disability rate, for example? Were most of your cases classified as multibacillary despite paucibacillary expected predominance among children? Is the coverage of household contact examination very low?

Authors: Due to limited study data, the authors cannot support the idea of underreporting cases of leprosy in children. The discussion of this theme was brought up based on other works carried out, however, in Imperatriz it is not possible to state that there is underreporting of cases (from the database used in this work). We try to raise a hypothesis on this topic, but we cannot state it.

The excerpt cited by the reviewer was deleted without affecting subsequent paragraphs.

Lines 419-430 – The important progressive increase in NCDRs for elderly females observed in your study from 2011 is really an intriguing result. This was the only rate to reach the same level from the beginning of your time series.

Your discussion addressed issues related to a possible later diagnosis for women than men, but you did not present any data from Imperatriz to support this hypothesis, for example, a recent change with increase of grade 2 disability rate for this sex and age group would be an indicator for that. We suggest a deeper investigation on possible specific actions taken for this sex and age group that could have increased leprosy detection from 2011.

Authors: We agree with the reviewer's comment, as we did not address in depth the specific actions taken for females in the age group of 60 years old or over that could increase detection from 2011.

Unfortunately, we have not found specific actions in the literature for women in this age group in the study setting or in the state of Maranhão.

One possible action that we problematize in the work concerns “The Enhanced Global Strategy for Further Reducing the Disease Burden Due to Leprosy (2011-2015)” which proposed the proportion of female leprosy cases indicator among the total of new cases indicator.

Although our study cannot measure the impact of this action, it is possible to observe that from 2011 to 2016 the trends are markedly increasing, a period coinciding with the effectiveness of the strategy (2011 - 2015).

Despite this hypothesis, further research is needed to understand why elderly women showed this markedly increasing trend in the study scenario.

The text was adequate:

Lines 483-493: “In Imperatriz, during the study period, no specific actions were found for women aged 60 years old or over that could explain the growing trend in this age group. The Enhanced Global Strategy for Further Reducing the Disease Burden Due to Leprosy (2011-2015) from WHO proposed the proportion of female leprosy cases indicator among the total of new cases, in order to assess and ensure that women are having adequate access to leprosy diagnostic services [54]. 

The implementation of this strategy and the creation of this indicator may have impacted the detection among women in the studied scenario, especially in women over 59 years old, considering that from 2011 onwards, the trend of this range shows an increasing behavior until 2016, concurrent with the period of validity of the strategy (2011-2015).”

Lines 437-440 – The text was repeated, please correct it.

In your whole study period NCDRs for children younger than 15 yo ranged roughly from 1 to 10 new cases per 100,000 children. On the other hand, you reported that NCDRs for the elderly fluctuated between 10 to 60 new cases per 100,000 people aged 60 or more. Why do you affirm that “individuals younger than 15 years old are more vulnerable to the disease”?

Authors: The repeated text has been corrected.

Regarding the statement “individuals younger than 15 years old are more vulnerable to the disease”, the authors consider this population to be epidemiologically important for leprosy control, since leprosy in children is an indication of active disease transmission, however, we cannot affirm that children under 15 are more vulnerable.

The excerpt was corrected, highlighting the importance of surveillance in children under 15 years old and in the group of 60 years old or over, as it was the two age groups that showed increasing trends during the study period:

Lines 494-504: “This study’s findings lead to a discussion regarding the profile of the disease in the context of the studied scenario, revealing that the trends found in this study period indicate that males under 15 years old and women aged 60 years old or over showed an increasing trend in the detection rate from 2006 to 2016. These findings indicate that health services should direct efforts to detect cases of leprosy actively, considering that having individuals younger 15 years old affected by the disease indicates active transmission within a household and/or social group. The results also indicate the need to maintain actions to diagnose the disease earlier in the population aged 60 years old or over especially because this age group presents the highest detections rates in the investigated scenario. Future studies should be carried out in order to understand why women have an increasing trend in detection in the age group above 59 years.”

Lines 440-442 – You did not report any data about physical disability among the elderly female.

Authors: The sentence mentioned was corrected, as we do not present and do not have data related to physical disabilities in elderly women:

Lines 500-504: The results also indicate the need to maintain actions to diagnose the disease earlier in the population aged 60 years old or over especially because this age group presents the highest detections rates in the investigated scenario. Future studies should be carried out in order to understand why women have an increasing trend in detection in the age group above 59 years.

Line 446 – please replace “early” by “earlier”.

Authors: The term “early” has been replaced by “earlier” (Line 504).

Lines 458-462 – Some published papers registered a real decrease of leprosy in some countries, such as Norway, where there were a decrease of transmission followed by a change in epidemiological pattern of disease towards older ages. This hypothesis should also be considered and discussed by authors in addition to their operational explanations for decreased chances of diagnosis.

Authors: The authors agree with the reviewer's comment, and discussed the decrease in transmission and change in the epidemiological profile of the disease in Imperatriz, using a study conducted in Norway as a reference for the discussion.

Our results show a decrease in total trends and by sex, as well as a decrease forecast. In addition to this decrease, our results showed high detection rates in the elderly (a result similar to the study carried out in Norway), which may be evidence of decreased disease transmission. It should be noted that, despite a likely decrease in transmission, the study scenario is hyper-endemic for the disease:

Lines 523-531: “In Norway, where leprosy was a serious public health problem in the 19th century, the reduction in transmission was accompanied by a change in the epidemiological profile of the disease, with a decrease in cases in young age groups and an increase in the proportion of elderly people among the new cases [35]. In our study, trends and forecasts of decreasing of total and by sex detection, accompanied by high detection rates in the age groups of 60 years old or older, may be indicative that leprosy transmission is decreasing. Despite this possible scenario of decreased leprosy and changes in the profile of patients, the state of Maranhão and the city of Imperatriz are hyperendemic for the disease.”

Lines 473-478 – We suggest a better definition of “elimination of leprosy as a public health problem” and of “elimination of transmission of leprosy”, which can be explained in the introduction of this paper. This will avoid misunderstanding of readers, especially at the end of the discussion section.

Authors: The definitions of “elimination of leprosy as a public health problem” and of “elimination of transmission of leprosy” were presented at the beginning of the introduction.

At the end of the discussion section, we support the conclusions of the paper on reducing the burden of leprosy (elimination of transmission of leprosy), showing that Imperatriz will not be able to reach the goals of reducing the burden of the disease for 2020.

We rewrote the passage pointed out by the reviewer to avoid misunderstandings among readers:

Lines 544-548: “The results of the adjusted models and trends show a decrease in the total detection rates, as well as in detection rates of men and women separately in the study period. However, considering the forecasts and trends, leprosy will remain endemic and the WHO global goals to decrease the disease’s burden and eliminate of transmission of leprosy by 2020 may not be met.”

Line 491 – we suggest reviewing your affirmation that there was a growing trend for women aged less than 15, despite the slight increase in 2016.

Authors: The passage indicated by the reviewer has been corrected. There was no growing trend in women under 15:

Lines 552-555: “In conclusion, the results show downward trends of detection rates, in total and by sex. Despite decreasing trends, growing trends were found in terms of age; men aged below 15 years old and women aged 60 years old or over presented increasing detection rates, which is relevant in terms of public policies and strategic actions.”

---

## [Decision Letter · Decision Letter 1]

2 Jun 2020

PONE-D-20-00967R1

Trends and forecasts of leprosy for a hyperendemic city from Brazil’s northeast: Evidence from an eleven-year time-series analysis

PLOS ONE

Dear Dr. Ramos,

Thank you for submitting your manuscript to PLOS ONE. After careful consideration, we feel that it has merit but does not fully meet PLOS ONE’s publication criteria as it currently stands. Therefore, we invite you to submit a revised version of the manuscript that addresses the points raised during the review process.

We look forward to receiving your revised manuscript.

Kind regards,

Eyal Oren, Ph.D.

Academic Editor

PLOS ONE

Reviewers' comments:

Reviewer's Responses to Questions

**Comments to the Author**

1. If the authors have adequately addressed your comments raised in a previous round of review and you feel that this manuscript is now acceptable for publication, you may indicate that here to bypass the “Comments to the Author” section, enter your conflict of interest statement in the “Confidential to Editor” section, and submit your "Accept" recommendation.

Reviewer #1: (No Response)

Reviewer #2: All comments have been addressed

2. Is the manuscript technically sound, and do the data support the conclusions?

Reviewer #1: Yes

Reviewer #2: Yes

3. Has the statistical analysis been performed appropriately and rigorously? 

Reviewer #1: Yes

Reviewer #2: Yes

4. Have the authors made all data underlying the findings in their manuscript fully available?

Reviewer #1: Yes

Reviewer #2: Yes

5. Is the manuscript presented in an intelligible fashion and written in standard English?

Reviewer #1: Yes

Reviewer #2: Yes

6. Review Comments to the Author

Reviewer #1: (No Response)

Reviewer #2: The manuscript reads much better now.

I would like to make few additional suggestions:

Abstract

Line 40 – I suggest using “average incidence” instead of “incidence” only

Line 48 – I suggest saying that ‘the city is unlikely to meet a "significant decrease" of the disease burden by 2020’

Introduction

Line 75 – I suggest using “In the same year” instead of “In the same period”

Materials and methods

Line 134 – “In the same period” means 2010? If so, it would be better to use “the same year”. The same should be checked for line 140.

Results

Line 237 – Revise the use of “both” once you report results for 3 groups, or rephrase it using two categories (for example: “total of cases and cases by sex”)

Discussion

Line 429 – Replace “by” for “be”

Lines 457-461 – Authors have just stated (lines 452-456) that primary health units do not present satisfactory performance in diagnosing leprosy in children because they don’t actively seek cases. Thus it is surprising that Anchieta’s study (2019) observed a decreasing trend in the detection of children under 15 years old because of active case-finding campaigns in the school-aged population. These two statements are not in agreement with each other.

Line 477 – We suggest to include “leprosy” before “patients”

Line 552 – Please replace “eliminate of transmission” by “eliminate the transmission”

7. PLOS authors have the option to publish the peer review history of their article (what does this mean?). If published, this will include your full peer review and any attached files.

Reviewer #1: Yes: Christine Murto, PhD

Reviewer #2: Yes: Mauricio Lisboa Nobre

---

## [Author Response · Author response to Decision Letter 1]

29 Jun 2020

Dear Editor, 

Many thanks for your reply and your reviewers' comments about our manuscript “PONE-D-20-00967 Trends and forecasts of leprosy for a hyperendemic city from Brazil’s northeast: Evidence from an eleven-year time-series analysis”.

The comments were appropriate to qualify/ improve the manuscript. We have forwarded a letter with the changes made in the manuscript, presenting our response for each comment from reviewers. Many thanks for your comments.

Kind regards,

Ramos et al.

Editor comments:

After final review, I find the manuscript has resolved most of the questions that were addressed to the authors. In the rewrite, there are some small edits that should be considered for revision: 

Line 63 change a to the

Authors: The term was included according to the suggestion.

Lines 61 – 64: “After the introduction of Multidrug Therapy (MDT) and the high vaccination coverage of the Bacillus Calmette-Guérin (BCG), especially in children, the burden of leprosy has decreased considerably worldwide”.

Line 68 remove : after are and remove ; and number

Authors: Terms have been revised.

Lines 67-72: “In 2016, the World Health Organization (WHO) published the Global Leprosy Strategy 2016−2020: Accelerating towards a leprosy-free world, the objectives are to decrease the disease’s global and local burden, decrease the cases of children with deformities, decrease the new cases diagnosed with grade 2 physical disabilities to less than one case per 1 million inhabitants, and review all laws that somehow lead to the discrimination against people with leprosy [4].”

Line 167 – do you mean causal and not casual? 

Authors: We mean causal. The sentence were corrected.

Lines 165-167: “Leprosy detection rates were smoothed by the moving average technique, considering the average of three months (prior, current and posterior), in order to remove noise and better reveal the underlying causal process”.

Line 292 change to high rates of disease were found

Authors: The sentence were corrected

Lines: 291-292: “Despite the downward trend seen in the age group between 30 and 59 years old, both among women and men, high rates of disease were found in the entire study period”.

Line 369 change most of the reported cases were male to largest proportion of cases were male

Authors: The sentence were corrected

Lines 368-370: “In this context, largest proportion of cases were male, which is in agreement with data provided in the literature, in which those affected by the disease in most world regions are predominately male, including in Brazil [27–29]”.

Line 374 punctually is not clear and I would suggest rewording

Authors: The word was reformulated according to the suggestion.

Lines 373-374: “In a few periods of the time series, women had slightly higher detection rates than men”.

Line 381 this is a long paragraph, suggest two sentences

Authors: The authors agree with the editor. The paragraph was divided into two sentences.

Lines 381-386: “Other factors may also explain the higher occurrence of the disease in men, as individual determinants, such as not seeking medical assistance or only later seeking health services, when compared to the practices of women. There are also operational issues concerning difficulties accessing health services in a timely manner due to an incompatibility between the men’s and the units’ working hours, a lack of a health policies directed to men, and restricted access to health information [15,16]”.

Line 389 I don’t see where transmission is increasing in men from the data, I’m not sure if I’m missing something. Transmission could be continuous because of maintenance of the disease but I don’t see where in the data you are indicating increases in this group

Authors: We agree with the comment. According to results it is not possible to confirm that transmission is increasing in men. We make it clear that from the results we have strong indications that transmission is greater in men.

Lines 387-389: “The higher detection rates in men compared to women, in the research scenario, especially in the period from 2011 to 2016, are strong indications that transmission is higher in men”.

Lines 397 and 401 change to year not years

Authors: The sentence was reformulated according to the suggestion.

396-403: “In terms of age, most of the individuals affected are aged between 30 and 59 years old, followed by the 15 to 29 year old group, though the population over 59 years old presents the highest detection rates. 

Despite the high rates in the 30 to 59 year age group, it should be noted that the temporal trends in these intervals were decreasing in the study period. And the 15 to 29 year old interval also showed the highest average percentage decrease in the period from 2006 to 2016, which points to an indication of a reduction in bacillus transmission in the studied region [32,35]”.

Line 404-407 this is a run on sentence, suggest revision

Authors: The sentence was revised.

Lines: 404-406: The group aged between 30 and 59 years old includes the Brazilian economically active population, interval that the disease hinders labor activities, forcing individuals to stop working or retire early, decreasing the quality of life of workers [28,36,37].

Line 408 remove “reaching”; change diagnoses to diagnosis of cases; in addition to… suggest to reword for clarity

Authors: The review was carried out according to the suggestions

Lines 406-410: “In this sense, health services should focus on preventive measures by actively seeking individuals in this age group, in addition to diagnosis of cases, providing timely treatment, and early identification of lesions, the purpose of which is also to prevent physical disability”.

Line 411 change “due to” to from

Authors: The sentence were corrected

Lines 410-412: “Early interventions prevent or minimize the high social costs leprosy imposes from removing this population from productive activities and social relationships [28,36-38]”.

Line 418 change to population and remove the apostrophe

Authors: The sentence were corrected.

Lines 417-421: “Considering the rapid aging process of the Brazilian and Imperatriz population, when leprosy is diagnosed and treated late, it leads to the functional loss of peripheral nerves and physical impairment, which, combined with the aging process and other comorbidities, contribute to elderly individuals’ greater vulnerability and loss of autonomy [41]”.

Line 424 and paragraph – this is not clear – you state that there is a continuous increase in the rate of detection that indicates a decrease in transmission. Are you intending to show that because the increase is in the older population and other ages show a decline, that this is an indicator in overall reduction? If so, you may want to revise. Although you do also show increases in <15 years among males. Alternatively you could remove the sentence since you have described the decreasing tendency in the following paragraph, and include with the exception of women 60 and older and males <15

Authors: In line 424 and in the paragraph we intended to show that the increase in the detection of elderly people may be indicative of a change in the epidemiological profile of the disease.

In some countries, the elimination of leprosy was followed by a change in the profile of the disease, with the greatest detections in the elderly.

Despite this change in the leprosy profile, Imperatriz has hyperendemicity of the disease.

We agree with the editor's suggestion and rewrite the paragraph, we try to make the paragraph more clearer. Thank you very much for the suggestion for improvement.

Lines 422-430: “Countries that registered a decrease in leprosy transmission, with subsequent elimination, observed a change in the profile of the disease, with a drop in detection in younger age groups and an increase in detection of elderly people [35,42]. In Imperatriz, the high detection rates in the population aged 60 years old or over may be an indicative of a change in the epidemiological profile of the disease, despite the municipality still showing levels of hyperendemicity.

A decreasing tendency was found in the total detection rates of both men and women from 2006 to 2016, however, considering the age groups, women aged 60 years old or over and men aged less than 15 years old showed an increasing trend”.

Line 438 include a period (.) after trends for a new sentence

Authors: Inclusion carried out.

Line 437: “This study’s findings show total and by-sex downward trends”.

Line 432 you discuss improvements in the decline of cases due to improved control efforts. However line 446 indicates that the process is weak. Perhaps reword to indicate that the process could be improved upon

Authors: According to the suggestion, we indicate on line 446 that the process can be improved.

Lines 444-446: “Potential explanations include difficulty establishing a clinical diagnosis, disease-related stigma, and the weak health promotion and education process, needing improvements in leprosy control actions in these areas [15,45,46]”.

Line 448 include For example at the start of the sentence. Change to an assessment of health services

Authors: The sentences were corrected according to the suggestion.

Lines 447-449: “For example, studies conducted in two Brazilian regions in which leprosy is endemic an assessment of health services identified that the local primary health units did not present satisfactory performance in diagnosing individuals younger than 15 years old”.

Line 451 reword “passively assisted voluntary demand” to health diagnosis was conducted on request, however active case detection in the community is not being conducted

Authors: The sentence was reformulated according to the suggestion.

Lines 450-451: “The reason for this is that the services health diagnosis was conducted on request, however active case detection in the community is not being conducted. [46,47]”.

Line 464 long sentence, suggest revision

Authors: The sentence has been reformulated. It was divided into two sentences.

Lines 463-468: Although the present study does not measure the impact of this action, the hypothesis arises that the National Campaign for Leprosy may be influencing the detection of cases under the age of 15 years old in the municipality, especially due to the rates found in the investigated period. It should be noted that the beginning of the campaign (2013) is concomitant with the beginning of the growth trend in men under 15 years old (2014)”.

Line 486 revise to proposed the inclusion of

Authors: The sentence was revised.

Lines 483-487: “The Enhanced Global Strategy for Further Reducing the Disease Burden Due to Leprosy (2011-2015) from WHO proposed the inclusion of female leprosy cases indicator among the total number of new cases, in order to assess and ensure that women are having adequate access to leprosy diagnostic services [54]”.

Line 487 revise to total number

Authors: Revised.

Lines 483-487: “The Enhanced Global Strategy for Further Reducing the Disease Burden Due to Leprosy (2011-2015) from WHO proposed the inclusion of female leprosy cases indicator among the total number of new cases, in order to assess and ensure that women are having adequate access to leprosy diagnostic services [54]”.

Line 549 do you mean database here?

Authors: Yes, it is a database. The term was corrected.

Lines 548-550: “This study’s limitations include the fact a secondary database was used, with inconsistent quality and quantity of information, with the potential presence of ignored or incomplete data”.

Reviewers' comments:

Reviewer #1: (No Response)

Reviewer #2: The manuscript reads much better now.

I would like to make few additional suggestions:

Abstract

Line 40 – I suggest using “average incidence” instead of “incidence” only

Authors: The term was included according to the suggestion.

Lines 39-42: “A total of 3,212 cases of leprosy were identified, the average incidence among men aged between 30 and 59 years old was 201.55/100,000 inhabitants and among women in the same age group was 135.28/100,000 inhabitants”.

Line 48 – I suggest saying that ‘the city is unlikely to meet a "significant decrease" of the disease burden by 2020’

Authors: The reviewer's suggestion was considered.

Lines 46-48: “Even though the forecasts show a downward trend in Imperatriz, the city is unlikely to meet a significant decrease of the disease burden by 2020”.

Introduction

Line 75 – I suggest using “In the same year” instead of “In the same period”

Authors: The term was included according to the suggestion.

Lines 75-77: “In the same year, Brazil presented a detection rate of new cases of 12.94 cases/100,000 inhabitants, accounting for 93% of the total cases reported in the Americas [5,6]”.

Materials and methods

Line 134 – “In the same period” means 2010? If so, it would be better to use “the same year”. The same should be checked for line 140.

Authors: The periods refers to 2010. We correct for "the same year".

Lines 134-140: “In the same year, the main social indicators are: an illiteracy rate of 9.7%, Human Development Index (HDI) of 0.73 and Gini Index of 0.46. In terms of basic sanitation, 23% of the city has a sewage system and 86% has a drinking water supply [18,19].

In 2016, the detection rate of new cases of leprosy in the state of Maranhão was 47.30 cases/100,000 inhabitants, classifying the state as the third most endemic in Brazil. In the same year, Imperatriz presented a detection rate of 62.23 cases/100,000 inhabitants, marking it as a Brazilian city with hyperendemicity levels [6,9,20]”.

Results

Line 237 – Revise the use of “both” once you report results for 3 groups, or rephrase it using two categories (for example: “total of cases and cases by sex”)

Authors: The excerpt was revised according to the suggestion.

Lines 235-238: “Table 1 presents the descriptive statistics of cases according to sex, age groups and AMPC, showing in absolute numbers, that the group aged between 30 and 59 years old predominated among total of cases (1566; rate=166.46/100,000 inhabitants), men (892; rate=201.55/100,000 inhabitants) and women (674; rate=135.28/100,000 inhabitants)”.

Discussion

Line 429 – Replace “by” for “be”

Authors: The term was replaced according to the suggestion.

Lines 424-427: “In Imperatriz, the high detection rates in the population aged 60 years old or over may be an indicative of a decrease in transmission, despite the municipality still showing levels of hyperendemicity”.

Lines 457-461 – Authors have just stated (lines 452-456) that primary health units do not present satisfactory performance in diagnosing leprosy in children because they don’t actively seek cases. Thus it is surprising that Anchieta’s study (2019) observed a decreasing trend in the detection of children under 15 years old because of active case-finding campaigns in the school-aged population. These two statements are not in agreement with each other.

Authors: The studies on lines 448-452 state that their scenarios did not perform satisfactory for the diagnosis of leprosy in children because local health units do not active case-finding. These studies [46,47] assess the performance of local health units. These studies were carried out before the year 2013, before the National Campaign for Leprosy, Vermin, Trachoma and Schistosomiasis

In the study by Anchieta, conducted in 2019, the consequences of national campaigns of active case-finding in school-aged children were discussed, which contributed to the drop in the trend of detecting children under 15 years of age.

We emphasize that the studies [46, 47] are local, and that Anchieta's study refers to national campaigns.

Lines 447-456: “For example, studies conducted in two Brazilian regions in which leprosy is endemic an assessment of health services identified that the local primary health units did not present satisfactory performance in diagnosing individuals younger than 15 years old. The reason for this is that the services health diagnosis was conducted on request, however active case detection in the community is not being conducted. [46,47]. 

Anchieta et al. (2019) [43] identified that Imperatriz showed a decreasing trend in the detection rate of children under 15 years old in the period from 2001 to 2015, a phenomenon explained by active case-finding national campaigns in the school-aged population in the years 2013 and 2016. Despite this decrease, the authors reaffirm that detection in children under 15 years old remains high in the municipality”. 

Line 477 – We suggest to include “leprosy” before “patients”

Authors: The term has been included.

Lines 471-474: “According to Monteiro et al. (2013) [36], 60.3% of the leprosy patients located in the north of Brazil were male individuals aged over 60 years old, while Nobre et al. (2017) [32] determined that 15.11% more men than women were affected by the disease”.

Line 552 – Please replace “eliminate of transmission” by “eliminate the transmission”

Authors: The term was replaced according to the suggestion.

Lines 545-547: However, considering the forecasts and trends, leprosy will remain endemic and the WHO global goals to decrease the disease’s burden and eliminate the transmission of leprosy by 2020 may not be met.

---

## [Decision Letter · Decision Letter 2]

22 Jul 2020

Trends and forecasts of leprosy for a hyperendemic city from Brazil’s northeast: Evidence from an eleven-year time-series analysis

PONE-D-20-00967R2

Dear Dr. Ramos,

We’re pleased to inform you that your manuscript has been judged scientifically suitable for publication and will be formally accepted for publication once it meets all outstanding technical requirements.

Kind regards,

Eyal Oren, Ph.D.

Academic Editor

PLOS ONE

Additional Editor Comments (optional):

Reviewers' comments:

Reviewer's Responses to Questions

**Comments to the Author**

1. If the authors have adequately addressed your comments raised in a previous round of review and you feel that this manuscript is now acceptable for publication, you may indicate that here to bypass the “Comments to the Author” section, enter your conflict of interest statement in the “Confidential to Editor” section, and submit your "Accept" recommendation.

Reviewer #1: All comments have been addressed

Reviewer #2: All comments have been addressed

2. Is the manuscript technically sound, and do the data support the conclusions?

Reviewer #1: Yes

Reviewer #2: Yes

3. Has the statistical analysis been performed appropriately and rigorously? 

Reviewer #1: Yes

Reviewer #2: Yes

4. Have the authors made all data underlying the findings in their manuscript fully available?

Reviewer #1: Yes

Reviewer #2: Yes

5. Is the manuscript presented in an intelligible fashion and written in standard English?

Reviewer #1: Yes

Reviewer #2: Yes

6. Review Comments to the Author

Reviewer #1: Very good and thank you for the revisions to the manuscript. A couple of very small grammatical revisions :

Remove "an" Lines 422-430: “Countries that registered a decrease in leprosy transmission, with subsequent elimination, observed a change in the profile of the disease, with a drop in detection in younger age groups and an increase in detection of elderly people [35,42]. In Imperatriz, the high detection rates in the population aged 60 years old or over may be an indicative of a change in the epidemiological profile of the disease, despite the municipality still showing levels of hyperendemicity.

Add a comma: Lines 447-449: “For example, studies conducted in two Brazilian regions in which leprosy is endemic, (add comma here) an assessment of health services identified that the local primary health units did not present satisfactory performance in diagnosing individuals younger than 15 years old”

I do not need to review again, these are fine upon correction.

Reviewer #2: The authors have solved all addressed issues and the manuscript reads much better. We believe their paper will help other researchers working with leprosy and should be published now.

7. PLOS authors have the option to publish the peer review history of their article (what does this mean?). If published, this will include your full peer review and any attached files.

Reviewer #1: No

Reviewer #2: **Yes: **MAURICIO LISBOA NOBRE

---

## [Editor Report · Acceptance letter]

28 Jul 2020

PONE-D-20-00967R2 

Trends and forecasts of leprosy for a hyperendemic city from Brazil’s northeast: Evidence from an eleven-year time-series analysis 

Dear Dr. Ramos:

I'm pleased to inform you that your manuscript has been deemed suitable for publication in PLOS ONE. Congratulations! Your manuscript is now with our production department. 

Kind regards, 

on behalf of

Dr. Eyal Oren 

Academic Editor

PLOS ONE